

# Local reduction of decadal glacier thickness loss through mass balance management in ski resorts

A. Fischer[1], K. Helfricht[1] and Martin Stocker-Waldhuber[1]

[1]Institute for Interdisciplinary Mountain Research, Austrian Academy of Sciences, Innsbruck, 6020, Austria

*Correspondence to*: Andrea Fischer (andrea.fischer@oeaw.ac.at)

**Abstract.** For Austrian glacier ski resorts, established in the 1970s and 1980s during a period of glacier advance, negative mass balances with resulting glacier area loss and decrease in surface elevation present an operational challenge. Glacier cover, snow farming and technical snow production were introduced as adaptation measures based on studies on the effect of these measures on energy and mass balance. After a decade of the application of the various measures, we studied the transition from the proven short-term effects to long-term effects by comparing elevation changes in areas with and without mass balance management. Based on LiDAR DEMs and DGPS measurements, decadal surface elevation changes in 16 locations with mass balance management were compared to those without measures (apart from piste grooming) in five Tyrolean ski resorts on seven glaciers. The comparison of surface elevation changes presents clear local differences in mass change, and it shows the potential to retain local ice thickness over one decade. Locally up to 20 m of ice thickness was preserved compared to non-maintained areas at glacier tongues over a period of nine years. At 11 out of 16 profiles with mass balance management measurements, surface elevation loss could be reduced by more than 35%. At six profiles, surface elevation loss was reduced by over 65%. At two of these profiles the surface elevation was preserved altogether, which is promising for a sustainable maintenance of the infrastructure at glacier ski resorts. Features like former covered pistes and installations in fun parks have rapidly evened out with the surrounding surface elevation as soon as maintenance was stopped. In general the results demonstrate the high potential of the combination of piste grooming and glacier cover, not only in the short term, but also for multi-year application to maintain the skiing infrastructure.

## 1 Introduction

During the last three decades, alpine glaciers have retreated drastically and increasingly, with current annual rates at historically unprecedented levels (Zemp et al., 2015). Globally, glacier mass balances have been increasingly negative (Vaughan et al., 2013), leading to additional river runoff in glacier-covered basins (Kovats et al. 2013). In Asia glacier retreat is likely to affect water scarcity (Hijioka et al. 2013).In the European Alps, glaciers are part of the national economy, contributing to hydropower production and as part of ski resorts. Glacier ski resorts are located at high elevations and thus are less affected by a decrease in depth height and duration of seasonal snow cover than lower ski resorts (Kovats, et al.,



2013). Recently, mass balance management methods have been developed to store snow in ski resorts (Skogsberg and Lundberg. 2005; Spandre et al., 2016; Grünewald and Wolfsperger, 2016) and manage meltwater production (Nestler et al., 2015; Norphel and Padma, 2015). In the Austrian Alps, mass balance management in glacier ski resorts started after the extreme melt in the summer of 2003 (Fischer et al., 2011) to compensate for the negative effects of glacier retreat on ski

resort infrastructure. This paper presents the long-term effects of the measures on local glacier elevation change.

Austria's glaciers experienced a reduction by 26% in area in recent decades (GI1 1969 – GI3 2006/2012; Fischer et al., 2015). Since the extreme summer of 2003, we have seen several years with negative mass balances in all elevation zones. The glacier changes of the last three decades were challenging for the eight Austrian glacier ski resorts (Table 1), which are located on 15 glaciers. They were opened between 1969 to 1987, when up to 72% of the Austrian glaciers were advancing

(Fischer et al., 2013b). During the early years of the glacier ski resorts, the main skiing season was during summer, with some of the resorts even being closed during winter. In recent decades there has been less demand for summer skiing  and the main season has shifted to autumn and spring. Most resorts open during summer for hikers and mountaineers only. Diolaiuti et al. (2006) investigated glacier evolution and summer skiing at Vadretta Piana (Stelvio Pass, Italy). The noticed that, although the glacier has receded, single years of exceptional good conditions for glacier summer skiing can still result

in a high number of skiers. A comparative study on the impact of glacier changes on mountain tourism was presented by (Smiraglia et al., 2008)

Not only the visitor demand developed over time but also cable car technology and with it the demands on glacier conditions. Initially it was mainly tow-lifts operating on the glacier, low installations with adjustable pylons to compensate for glacier flow and mass balance. As these lifts transport the skiers along the ground, they are technically easier to maintain

and have less strict corridors for compensating glacier motion and mass balance. However, tow-lifts need a route with a gentle slope. Nowadays, chair lifts and circulating ropeways are built with much higher pylons and bearing loads. While these lift types can also be built in complex terrain of steep slopes or rock cliffs, there are strict limits on the acceptable inclination of the pylons. Apart from lift infrastructure, pistes on the glacier surface have to fulfil specific requirements regarding width and steepness. The transition of the ski tracks from glacier to the bare ground changes constantly with

variations in glacier surface altitude and snout position.

The loss of firn reservoirs, increase of debris on the glacier surface by melt and rock falls (Fischer, 2010) as well as more and deeper supraglacial channels increase surface roughness on glaciers, so that more snow is needed in grooming to smooth pistes (Fischer et al., 2011a). Where glacier ice has disappeared, bare ground is often steeper than and not as smooth as the former glacier surface, so that pistes have had to be rerouted to meet the requirements on width and difficulty. Sinking

glacier surfaces often make exit and entrance to summit or valley lift stations difficult. Steeper glacier surfaces complicate the maintenance of traverse pistes and increase the danger of avalanches. As the ropeway pylons are mounted on sledges designed for specific pylon shifts, changes in the flow regime, i.e. velocity and/or direction shorten maintenance windows. In

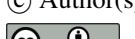



the worst case, sinking surfaces lead to angles and bearing loads which are out of the approved range for that installation, so that the ropeway has to be replaced or adapted. One positive effect of the years with negative mass balances was the decrease of ice flow velocities which led to a reduction of the number and size of crevasses in glacier ski resorts (e.g. Fischer et al., 2011a; Diolaiuti et al., 2006).

Therefore, mass balance management in glacier ski resorts has three aims:

i)        Decrease surface roughness by keeping snow over the summer

ii)       Keeping surface elevation around infrastructure

iii)      Prevent or reduce ice melt to keep bedrock ice-covered.

In previous studies at the glacier ski resorts in Tyrol (Austria), later also at Dachstein Glacier ski resort, several methods for
mass balance management in glacier ski resorts were investigated by extensive field work and modelling (Olefs and Fischer, 2008; Olefs and Obleitner, 2007; Olefs and Lehning, 2010):

-        Glacier covers

-        Grooming

-        Water injection

-        Snow-farming

Glacier covering means insulating the glacier surface with an approx. 2 mm thick white polypropylene fabric in the period between peak accumulation (mid-May) and the start of the accumulation season (early September). Piste grooming comprises regular mechanical preparation of the ski piste by snow cat during operation (i.e. between September and May). Water injection aims at the infiltration and refreezing of liquid water in the snow layer to increase density and cold content.
Snow-farming summarizes efforts to amass snow accumulated from wind drift, technically produced snow and snow from avalanche deposits, which is relocated by snow cats to create snow depots or increase accumulation on the piste.

In the study of Olefs and Fischer (2008), glacier cover was by far the most effective method and reduced ablation by 60%. In case enough snow was accumulated during the winter or brought in with snow cats or wind drift, local annual mass balance even went positive during the experiments. Grooming without other measures reduced ice ablation by 10% by limiting wind
erosion of snow (Fischer et al., 2011a). Fahey et al. (1999) observed up to 45% more water available on groomed pistes compared to non-groomed slopes. The application of water injection into the snowpack was not developed further. It

increases the mechanical resistance of the piste, but has little effect on local glacier mass balance (Olefs and Fischer, 2008). As all measures are costly and need much manpower, application is limited to small areas, which have been identified as areas where sinking surface elevation, bare ground or steep slopes would do the most harm to the infrastructure. Thus mass balance management is applied only on less than 10% of the ski resorts glacier area, with mean values of about 3%, limiting the impact of the measures to hydrology and total glacier mass balance. As an additional method, snow production facilities have by now been installed at a number of ski resorts, providing snow on pistes for an early season start even on bare ice surfaces, when firn cover is missing, and to reduce ice ablation in summer.

After a decade of measuring the glaciers, the question arises of the long-term outcome of these measures: Although the short-term effect has been proven, it could be that measures have not been applied frequently enough to return a sustainable result, or that ice dynamics lead to a redistribution of masses so that, for example, no effect on surface elevations would be measurable. From this basic research question, this study aimed at assessing long-term net effects by comparing surface elevation changes in areas which have been subject to different types of mass balance management and neighbouring areas without such management. The investigated measures were accumulation by snow production and movement of snow with snow cats in combination with glacier covers. Surface elevation changes were compared to neighbouring areas of the same elevation and exposure without such measures (apart from some grooming of pistes).

## 2 Data and methods

### 2.1 Surface elevation data

The digital elevation models (DEMs) which are part of the 2nd and 3rd Austrian glacier inventory (Fischer et al., 2015; Lambrecht and Kuhn, 2007) are based on ortho-photogrammetry for GI2 (5 m grid size) and airborne laser scanning (ALS) surveys for GI3. For some of the ski resorts, DEMs exist from more recent ALS surveys (1 m grid size). Additionally, we performed DGPS measurements of the surface elevation along profiles using a TOPCON HiPer V Dual-Frequency GNSS Receiver. Raw DGPS data were corrected in post-processing with data of the reference stations Merano, Bolzano, Vipiteno and Malles Venosta provided by the autonomous province of South Tyrol / Alto Adige (http://www.stpos.it/SpiderWeb/frmIndex.aspx). All elevation data are summarized in Table 2.

### 2.2 Surface elevation changes



Local glacier thickness changes (Δh) result from mass changes caused by surface or basal mass balance (Δm), from the densification of snow and firn (Δρ), and from changes in submergence or emergence resulting from ice flow (Δv) (Cogley et al., 2011).

5    $\Delta h = \Delta m/\rho + h\cdot\Delta\rho + \Delta v$                                                 (1)

Assuming that the mean density (ρ) of the entire column containing ice, snow and firn is constant (Δρ = 0) and glacier thickness changes (Δh) are similar to surface elevation changes (Δz) caused by mass changes and changes in ice flow between two dates of surveys t0 and t1:

 $h_{t1}-h_{t0} = z_{t1} - z_{t0} = (m_{t1}-m_{t0})/\rho + ((v_{t1}-v_{t0})/2)\cdot(t1-t0)$                    (2)

In the ablation area, thickness change can be positive if ablation decreases between t0 and t1 and/or emergence velocity increases. In the accumulation area, positive thickness changes occur when accumulation increases and/or submergence

15   velocity decreases.

The geodetic surface elevation changes are calculated by subtracting the altitude at the dates of the first survey (t0) and of a second survey (t1) at every DGPS measured point of the profile

 $\Delta z = z_{t1} - z_{t0}$                                                         (3)

In case two DEMs were used, the elevation changes were calculated for the equidistant nodes (spacing =1 m) along a profile line.

The DGPS-measured altitude represents the altitude of the measurement point within the horizontal measurement uncertainty which was better than 0.2 m after post-processing. The DEM altitude represents the mean altitude within the pixel located at

25   the DGPS measurement point. Seasonal snow cover was neglected.

## 2.3 The study sites

The investigated glaciers where chosen with respect to data availability from previous studies in these glacier ski resorts, i.e. Stubai Glacier, Pitztal Glacier, Kaunertal Glacier and Sölden Glacier ski resort (Tab. 1). In addition to these ski resorts,

30   Hintertux Glacier was chosen because this resort has one of the first on-glacier snow production facilities. Within the resorts we identified areas with long-term mass balance management (Figure 1). Examples of such areas are presented in the pictures of Figure 2.



For profiles in these areas, altitude changes in neighbouring profiles with and without snow and mass balance management were compared. All measured profiles are subject to grooming during winter. During summer, only the Hintertux Glacier ski resort operates summer skiing and grooms the pistes. In addition to that, the glacier is covered during summer and/or technical snow is produced in winter at the profiles, but not at the reference profiles which are located nearby in similar

settings in terms of slope, aspect, shade and snow accumulation. In the **Hintertux Glacier ski resort** on the Gefrorene Wand Kees (Tuxer Ferner), surface elevation changes for the periods 1999 to 2007 and 2007 to 2015 were analysed for individual parts of a DGPS profile. The profile HI1 (Fig. 3) follows a piste, which is groomed with technical snow and partly covered with sheets in summer. The lowest points in the profile belong to an off-piste traverse towards another piste. Profile HI2 (Fig. 4) follows a piste across a flat area, where the glacier ice is covered in summer. Profiles HI3 and HI4 (Fig. 5 and 6)

cross a piste from the off-piste glacier surface towards the glacier margin. In this area a combination of technical snow production and glacier cover in summer is applied to reduce ablation at the glacier margin. Profile HI5 (Fig. 7) is located on a small glacier tongue, which is covered.

For **Stubai Glacier ski resort**, the latest available DEM for the glacier ski resort dates from 2006. In 2015, DGPS

measurements were performed at locations where different measures have been used since 2004.

Profile ST1 (Fig. 8) is located along a ski traverse from the cable car mountain station to the glacier col. This traverse is necessary to exit from the lift station. Ongoing reduction in glacier surface elevation and thus steeper slopes or even melt out of rocks between the station and the piste on the glacier would disconnect those two. The traverse is packed with additional snow and subsequently covered with sheets every summer. Profile ST2 (Fig. 9) is located on an ice divide which connects

the two glaciers Gaißkarferner and Windacher Ferner. Profile ST3 (Fig. 10) comprises an area next to a mountain lift station. In this area surface elevation is preserved to maintain the gentle slope at the start of the piste. To receive reference data, the analysed profile was extended further down the piste.

A very local application of the glacier cover is presented in profile ST4 (Fig. 11). The base of a pylon of the cable car has been covered every summer since 2008, when the pylons were last replaced to compensate for the ice flow. Reference data

were collected around this spot and along a profile on the piste nearby.

Data of profile ST5 (Fig. 12) mark an area essential for the operation of the Stubai Glacier ski resort. The cable car station is built on a rocky outcrop which originally divided the ice flow of Schaufelferner. The glacier surface elevation next to the cable car station was decreasing so that skiers would have to walk up to the station. The area has been covered every summer since 2004, with additional snow being deposited there. Reference data were collected from the piste above this location and

30 from an area to the side of this location.

In the **Sölden glacier ski resort**, profiles SOE1 and SOE2 (Fig. 13) show elevation changes along a piste on Rettenbachferner. In the area around profile SOE1, technical snow is added to the natural snow cover, whereas along SOE2 the influence of these technical snow loads is reduced.

At profile SOE1a (Fig. 14), avalanche snow is moved from the periglacial area onto the glacier tongue. The area has been covered every summer for 12 years to maintain the piste connection from the glacier to the valley station.

Profile SOE3 (Fig. 15) is a length profile located on Tiefenbachferner. Detailed DGPS surveys were performed directly on this glacier tongue in 2015 (SOE3a, Fig. 16). Profile SOE4 (Fig. 17) presents data from a traverse in the upper part of Tiefenbachferner.

As an up-to-date DEM is available, no additional DGPS measurements were performed in the **Pitztal glacier ski resort**. Profiles PI1 and PI2 (Fig. 18) present surface elevation changes on a tow lift route compared to the piste area. Profile PI3 (Fig. 19) is the corresponding cross section in this area.

In Pitztal Glacier ski resort profile PI4 (Fig. 20), mass balance management was applied until 2006 to preserve a tow lift track and a piste. As the cable car was rebuilt as a detachable gondola lift on a new ice-free route and the ski piste relocated, no measures were applied after 2006. Profile PI5 (Fig. 21) shows the maintenance of the piste in the highest parts of Mittelbergferner.

In the **Kaunertal Glacier ski resort** profiles KT1 and KT2 (Fig. 22) present the evolution of huge kickers built in a fun park on Weissseeferner. Profile KT3 (Fig. 23) was recorded at a huge snowboarder bump with an ice cave inside. This bump was created by amassing snow and covering it with sheets during summer. Profile KT4 (Fig. 24) presents a ski traverse in the upper part of Weissseeferner. DGPS data of profile KT5 (Fig. 25) were collected along a tow lift track.

## 3 Results

### 3.1 General Results

In general, in the last decade a balance of the surface elevation at high elevations could be achieved by snow grooming and by covering the glacier at the highest elevations (e.g. profiles HI1, ST4). The spatial heterogeneity of surface elevation changes was levelled out by using glacier cover sheets (e.g. ST2). At the glacier tongues the body of ice could be protected against total mass loss by this method (e.g. HI5, ST5, SOE1a, SOE3a). However, small features like artificial bumps and kickers do not have any long-term influence on glacier surface elevation, because they disappear within a short time (e.g. ST4, PI5, KT1, KT2). Where snow is gathered for piste maintenance, mass gain on the piste is balanced with mass loss in the areas where the snow is taken from. (e.g. SOE4, KT4). At 11 of the 16 profiles with mass balance management measurements, surface elevation loss could be reduced by more than 35% (Tab. 2). At 6 profiles surface elevation loss could even be reduced by more than 65%. At two of these profiles the surface elevation could be preserved altogether (ST1, HI4).



### 3.2 Hintertux Glacier ski resort

Profile HI1 shows a constant surface elevation along the piste with mass balance management (*mbm*) within the last 4 years (Fig. 3). Prior to that, the surface elevation changes were negative from 1999 to 2011 in the same profile. In the lowermost parts of this profile, the traverse between the two pistes lost a total of 8 m in surface elevation over the second period.

Profile HI2 shows a higher surface elevation loss in the reference areas (*ref*) compared to the middle part of this profile, which is covered in summer (Fig. 4). However, this local reduction is also visible in the data of the first period, but more pronounced in the second period with up to 4 m. The surface elevation changes in the second period were higher compared to surface elevation changes in the longer 12-year first period. A distinct reduction of glacier surface elevation loss is achieved by technical snow production along the glacier margin in the lower part of Gefrorene Wand Kees. Profiles HI3 and HI4

present mean differences in surface elevation changes of 5 m between the reference and the mass balance management areas for the second period, whereas surface elevation changes were within 2 m of difference between *mbm* and *ref* areas (Fig. 5 and 6, Tab. 2). Surface elevation of the 2011 surface could be kept constant at the glacier tongue over the second period (HI5, Fig. 7).

### 3.3 Stubai Glacier ski resort

The differences between *mbm* parts and *ref* parts of profile ST1 show that maintenance of the traverse started before 2006 (Fig. 8, Tab. 2). In both periods, surface elevation changes were more negative in *ref* parts compared to surface elevation changes in *mbm* parts, while the difference between them stayed nearly similar for both periods. Some surface elevation gain can be seen in the *mbm* part of profile ST1 in the second period.

Mean surface elevations changes are almost equal at both parts of the profile (Fig. 9). However, the variation of surface elevation changes is distinctly reduced in the second period. The standard deviation of surface elevation changes (Tab. 2) is reduced from 2.9 m in the first period to values below 1 m in the second period.

In the area of profile ST3 a positive surface elevation change can be observed in the second period, whereas surface elevation was rather constant in the first period (Fig. 10). However, surface elevation changes of the second period come

close to the surface elevation changes of the first period following the profile down-glacier along the piste.

The very local application of glacier cover to preserve the ice body for the pylon of the ropeway in profile ST4 caused differences of up to 9 m in surface elevation (interquartile range of *mbm* p2, Fig. 11). The achieved mean reduction in surface elevation around the pylon (compared to the piste nearby) was 2.5 m in (Tab. 2).

The area of profile ST5 presents some mass gain as a consequence of constant piste grooming and glacier cover (Fig. 12). In

mean the surface elevation in the mbm area could almost be preserved in the second period, whereas in the ref area surface elevation loss of the second period was in the same magnitude as in the first period.



### 3.4 Sölden Glacier ski resort

The continuous grooming of the piste on Rettenbachferner with technical snow production resulted in distinct differences in surface elevation changes of profiles SOE1 and SOE2 in the second period, whereas mean surface elevation changes were nearly the same between 1997 and 2006 (Fig. 13, Tab. 2). Between 2006 and 2014, mean surface elevation change at profile

SOE1 was nearly half that of profile SOE2 (Tab. 2). Locally more than 10 m of glacier surface elevation could be preserved in profile SOE1a (Fig. 14) using piste grooming and glacier cover at the glacier tongue.

On Tiefenbachferner, surface elevation losses were reduced locally on the glacier tongue. This is obvious at the lowest parts of profile SOE3 (Fig. 15) and in more detail in profile SOE3a (Fig. 16). Mean surface elevation loss was reduced by 65% (Fig. 16, Tab. 2). Profile SOE4 presents a large variability of surface elevation changes along a ski traverse on a slope. Less

elevation change can be found on the traverse compared to surface elevation changes above and below it (Fig. 17). However, mean surface elevation changes of the two periods are approx. similar.

### 3.5 Pitztal Glacier ski resort

Both PI1 and PI2 present less negative surface elevation change in the second period than in the first period (Fig. 18). As a result of covering the glacier rim to maintain the lift traverse along PI1, surface elevation changes in the lower part of this profile were distinctly less compared to those in PI2 at the same elevations. A small dislocation of the tow lift line in PI3 caused large surface elevation changes at its south-facing slope (Fig. 19). Also the dominance of the lift traverse compared to the glacier surface was more pronounced in 2006. The overdeepening next to the rocks in the southern part of PI3 appeared

in the second period (Fig. 19). Despite the high elevation location of profile PI4 in the accumulation area, changes are obvious in the position of the wind kolk (Fig. 20). The retreat of the edge of the piste caused high surface elevation losses at the old slope. Whereas these surface elevation changes in the second period were in the magnitude of those of the first period, the piste grooming and amassing of snow reduced surface elevation changes compared to the unprepared slope in a mostly snow-covered area. In profile PI5 the technical maintenance of a piste (western part) and a lift traverse (eastern part)

between 1997 and 2006 as well as the abandonment of this maintenance are obvious (Fig. 21). The technical features on the glacier surface in 2006 were completely reduced to the surrounding surface in 2014.

### 3.6 Kaunertal Glacier ski resort

Profile KT1 presents the construction of large kickers in a ski fun park before 2006 and the decay of these features after 2006

(Fig. 22). Profile KT2 shows mean surface elevation changes similar to those of KT1 in the second period, with the only difference that the kickers were reduced in KT1 and newly constructed in KT2. Both profiles show higher surface elevation



loss in the second period. The top of the bump in profile KT3 was built after 2006 and still reaches the original surface elevation of 1997 (Fig. 23), whereas surface elevation decreased sharply above and below the bump in the second period. This may have been caused by the use of the surrounding snow for the bump itself, which increases ice melt in these areas.

## 4 Discussion

The uncertainty of surface elevation changes stems from the uncertainty of the DEMs and the uncertainty of the DGPS measurements. DEMs from stereo-photogrammetric surveys have an uncertainty of surface elevation of 0.5 m, whereas DEMs from Airborne Laser Scanning (ALS) surveys have higher accuracy and thus lower uncertainty (e.g. Abermann et al., 2010). Especially on glacier surfaces, slopes are gentle and thus uncertainty of ALS elevations is small, with typical values of less than 0.2 m (e.g. Bollmann et al.,2011; Joerg et al., 2012, Deems et al., 2013). The uncertainty of the vertical
component of the DGPS location is assumed to be 1 m (e.g. Monteiro et al., 2005). For the difference between the DEMs and DGPS data, a maximum uncertainty of 1.1 m can be calculated. Seasonal snow cover was neglected. Especially at high-elevation profiles, the state of the snowpack may influence the results. For instance, the mean surface elevation change in the *mbm* part of profile ST1 of 2.2 m can partly be explained by the seasonal snow pack under the glacier cover at the time of measurement. The LiDAR DEMs were recorded in late August and September. The DGPS surveys took place in July and
August.

This study focuses on the analysis of glacier surface elevation changes, as these are a major challenge for the ski resorts. The interpretation of surface elevation changes in terms of mass balance is not possible without additional information. Potential sources of uncertainty on a local and glacier-wide scale are subglacial erosion, internal and basal melt and density changes (Cogley et al., 2011). These factors are usually neglected in geodetic mass balance studies, and we have no indication that
they would play a major role in our study. Much more importantly, according to eq. 2, surface elevation changes result from submergence and emergence and mass balance. In the ablation area emergence reduces surface elevation loss by ablation, so that ablation generally is higher than surface elevation decrease. In the accumulation area, accumulation is higher than surface elevation change as submergence takes place. Horizontal ice flow velocity on Austrian glaciers generally decreased (Fischer, 2015) and so did submergence and emergence (e.g. Span and Kuhn; 2003, Fischer et al., 2011b; Helfricht et al.,
2014). Interannual differences in emergence and submergence are less than 0.5 ma$^{-1}$ on Kesselwandferner (Fischer et al., 2011b). In any case, submergence and emergence should be similar for the profiles and the reference profiles. The shape of elevation changes in the DEM differences fits the location of the measures, so that a large impact resulting from different or changing ice flow regimes is unlikely.

However, this study does not focus on the absolute values of surface elevation changes. The aim is to analyse the differences between maintained glacier areas and areas with limited maintenance nearby. In the profile plots all measurement points are shown along the elevation range of the profile, so that differences caused by the measurement setup are obvious. Some of the



reference surfaces are subject to grooming, some are not. Local mass balance measurements indicate that grooming without other measures reduces ice ablation by 10% by limiting the wind erosion of snow (Fischer et al., 2011a). Relocation of snow by snow cats is mainly taking advantage of periglacial snow or even, in addition to that, deposits gained by blasting.

In general the absolute values of the mean surface elevation changes strongly depend on the chosen path or profile line and do not represent glacier mass balance at these elevations. Often the basis of pylons and lift traverses are covered to retain them over a period of several years, until the pylons have to be relocated to compensate for the ice flow. After stopping the mass balance management, these features, at first standing proud from the glacier surface, disappear fast. This can be explained by the enlarged surface of the feature in relation to its volume. Thus the increase of energy exchange will cause

higher melt until the surface is minimized and evens out with the nearby surfaces. Additionally, less snow is accumulated on it, because the surface is more exposed to wind.

In general it is not feasible to reconstruct the snow volume onto the piste. The locations of multi-year use of glacier cover are better known. Thus these areas where selected for the mbm areas along the DGPS profiles (blue colour, e.g. ZT1-5). However, these areas may also change slightly over the years so that the transition between maintained and non-maintained

areas is more fluid than the manual separation in mbm parts and ref parts shows. In addition to the geodetic analysis, GPR measurements have been carried to out to find the transition from new firn to the glacier ice at the glacier tongues. GPR data were recorded with a 500 MHz antenna to complement the DGPS measurements. However, while the glacier bed could be detected in most of the data, internal layering was not visible. This can be caused by the reduced differences in snow and firn layering caused by steady piste grooming, or by a lack of layers with low density.

Mass balance management at the glacier tongues may be feasible as long as the area to be managed remains small and needs not to be extended to larger areas upglacier. However, mass balance management shows the potential to keep the surface constant at highest elevations of the glaciers and thus conserve the firn reservoirs. This might have a long-term impact on the future existence since the natural glacier ELA in recent years often exceed peak elevation (Fischer et al., 2013a; Fischer et

al., 2014a; Fischer et al., 2014b). Thus, specific mass balance management in the typical firn areas is more sustainable with respect to future glacier extent than mass balance management at the tongues.

 Apart from the effects on mass balance, the economic benefit of mass balance management is often discussed, as well as the sustainability of measures in the light of current glacier retreat.

The economic benefit results from costs and gains, with costs for all investments being easier to capture than the gains. The

total costs of glacier covers are those of material and maintenance. Material/investment costs include sheets and bags filled with gravel for fixing the sheets on the glacier, and storage space. Maintenance costs include transport, mounting, maintenance on the glacier and removal of the material, both personnel and machinery costs. Depending on individual settings, total costs are about 1.5 €/m², divided about 50:50 between material and maintenance. The uncertainty about the economic benefits is much higher, as, even with detailed visitor questionnaires, the costs of loss of glacier area for ski slopes



is hard to quantify. In addition to that, the costs of the loss of glacier area or altitude are highly individual: If a ski lift has to be rebuilt, economic costs of glacier loss are quite high.

Another fact to keep in mind is the sustainability of measures on glaciers: Glaciers are constantly changing, so that some maintenance effort is always needed for adapting to retreat or, as was the case in the 1980s, to advance. Taking into account

that snow cover duration is high in today's glacier-covered regions, ski tourism 2100 might focus on these high-altitude regions, even if no glacier at all was left by then. The history of ski tourism is not very old. It started about 1900 and boomed in the 1970s in terms of infrastructure and turnover. In the light of changing markets, demands and politics, the climatic changes might introduce fewer uncertainties than the socioeconomic ones. In general, investments and facilities are budgeted for a time frame of than less than 20 years.

A wider application of the methods for meltwater management has been proven for Armenia (Nestler et al., 2014). An application in high lying regions in Central Asia could be feasible: Albedo has been shown to be a major factor governing mass balance (Fujita and Ageta, 2000), so that the application of geotextiles will reduce melt in the absence of seasonal snow falls. As the covers can be placed and removed at nearly any time (unless superimposed ice forms on them), an effective water management seems possible. Drawbacks of the method are the need for machinery for an application on areas larger

than about 100 x 100 m, and the costs.

An application of geotextile covers to ski resorts at lower elevations without glacier cover is not straightforward, as the sensible rather than the radiative energy flux is decisive here. This makes mass balance management by relocation of snow often combined with insulating measured as wood chips more effective than the albedo increase by geotextiles.

## 5 Conclusions

The use of snow grooming and technical snow production as well as glacier covering in selected areas on glaciers, which are important for the infrastructure and the pistes in glacier ski resorts, show good results in preserving the surface elevation on the decadal time scale. Distinct differences between surface elevation changes in maintained areas and surface elevation changes in nearby areas without technical intervention are presented in this study. Small-scale ice ridges arising from very local mass balance management melt down within a few seasons when mass balance management is stopped.

Up to now the areas under mass balance management represent only a small proportion of the total glacier area and thus have limited influence on the mass balance of the total glacier. Surface elevation differences between maintained and not technically prepared areas on the glaciers can be expected to increase with ongoing glacier retreat, which will cause steeper slopes on the glacier surface. In the uppermost parts of the glaciers the preservation of surface elevation by covering the glacier works well to retain the piste connection between ropeway mountain stations and the glacier surface over multi-year

periods. The long-term use of glacier cover in the upper parts of the glaciers (e.g. ST1, ST2) may affect the existence of these glacier parts in future, because equilibrium line altitudes of glacier mass balances in recent years have exceeded peak





elevations. In areas near the glacier terminus, the continuous combination of additional snow load and glacier cover helps to preserve the remaining ice body where, without mass balance management, the glacier would retreat rapidly.

Over the observed time periods, the reduction in surface elevation caused by glacier retreat could be reduced locally by more than 75%. Mass balance management measures thus do a good job in stemming surface elevation decrease on a small proportion of the area of ski resort glaciers where the measures can be applied. The application is limited by the effort necessary as well as by the limited snow and water resources.

**Acknowledgements**

The authors thank the Austrian glacier ski resorts for the constructive cooperation. We are grateful to Brigitte Scott for the English editing.

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



Table 1: Glacier ski resorts in Austria with opening year, federal state, glacier names, the total glacier area, the glacier area assigned to the ski resort and the relative area of the ski resort on glacier with mass balance management (mbm). An asterisk (*) denotes ski resorts with preparatory studies to mass balance management measures. A plus sign (+) denotes ski resorts with data analysed in this study.

| Ski resort | Opening year | State | Glaciers | Total glacier area (km²) | Ski resort area on glacier (km²) | Area of mbm (%) |
|---|---|---|---|---|---|---|
| Kitzsteinhorn | 1965 | Salzburg | Schmiedinger Kees | 1.16 | - | - |
| Dachstein Glacier * | 1969 | Upper Austria | Schladminger Gletscher | 0.71 | - | - |
| Hintertux Glacier [+] | 1969 | Tyrol | Gefrorene Wand Kees, Riepenkees | 4.56 | 4.56 | 2.9 |
| Stubai Glacier *[+] | 1972 | Tyrol | Schaufelferner, Daunferner, Fernauferner, Windacher Ferner, Gaißkarferner | 4.48 | 4.10 | 2.4 |
| Sölden *[+] | 1975 | Tyrol | Rettenbachferner, Tiefenbachferner | 2.76 | 2.76 | 2.2 |
| Pitztal Glacier *[+] | 1983 | Tyrol | Mittelbergferner, Brunnenkogelferner | 10.94 | 3.39 | 2.1 |
| Kaunertal Glacier *[+] | 1980 | Tyrol | Weisseeferner | 2.64 | 2.13 | 6.6 |
| Mölltal Glacier | 1986 | Carinthia | Wurten Kees | 0.05 | - | - |
| | | | total | 25.43 | 16.94 | 2.95 |

Table 2: Dates of the surface elevation information in the different glacier ski resorts from digital elevation models (DEM) based on orthophotos (O), airborne laser scanning surveys (ALS) and from differential GPS measurements (DGPS).

| Ski resort | Period 1 | | Period 2 | |
|---|---|---|---|---|
| | DEM GI2 (O) | DEM GI3 (ALS) | DEM (ALS) | DGPS |
| Hintertux Glacier | 1999 | 2007 | - | 03/08/2015 |
| Stubai Glacier | 1997 | 2006 | - | 06/07/2015 |
| Sölden Glacier | 1997 | 2006 | 2014 | 16/07/2015 |
| Kaunertal Glacier | 1997 | 2006 | 2012 | 23/07/2015 |
| Pitztal Glacier | 1997 | 2006 | 2014 | - |




Table 3. Mean (µ) and standard deviation (σ) of surface elevation changes [m] for areas of mass balance management (mbm) and no mass balance management (ref) at the profiles in the two periods (dates see Table1). Absolute difference (abs. diff, [m]) of the *mbm* mean value to the *ref* mean value and relative reduction (rel. red, [%]) in surface elevation change are given for the two periods.

| Profile | Period 1 | | | | | | Period 2 | | | | | |
| --- | --- | --- | --- | --- | --- | --- | --- | --- | --- | --- | --- | --- |
| | ref | | mbm | | abs. | rel | ref | | mbm | | abs. | rel. |
| | µ | σ | µ | σ | diff | red | µ | σ | µ | σ | diff | red |
| HI1 | -1.3 | 2.8 | -2.9 | 3.1 | -1.6 | 127 | -2.4 | 2.5 | -0.3 | 1.5 | 2.0 | -86 |
| HI2 | -4.2 | 0.5 | -3.1 | 0.4 | 1.1 | -26 | -5.3 | 1.3 | -4.3 | 0.9 | 1.1 | -20 |
| HI3 | -6.0 | 2.1 | -4.2 | 1.3 | 1.7 | -29 | -5.8 | 1.1 | -0.9 | 2.5 | 4.9 | -85 |
| HI4 | -4.2 | 1.4 | -5.7 | 1.2 | -1.4 | 34 | -6.0 | 1.1 | 0.1 | 1.1 | 6.1 | -101 |
| HI5 | -9.2 | 4.2 | -8.9 | 3.2 | 0.3 | -3 | -4.4 | 4.7 | -2.8 | 2.5 | 1.6 | -37 |
| ST1 | -7.4 | 3.8 | -0.7 | 3.1 | 6.7 | -91 | -4.7 | 2.4 | 2.2 | 1.9 | 6.9 | -148 |
| ST2 | 0.2 | 2.9 | -0.7 | 2.9 | -0.9 | -396 | 1.2 | 0.4 | 1.6 | 0.8 | 0.3 | 28 |
| ST3 | -3.6 | 0.4 | -0.2 | 2.0 | 3.5 | -95 | 0.5 | 2.1 | 4.2 | 1.9 | 3.8 | 828 |
| ST4 | -5.8 | 0.8 | -4.9 | 0.6 | 0.9 | -15 | -6.3 | 1.4 | -3.8 | 1.8 | 2.5 | -39 |
| ST5 | -9.8 | 1.0 | -8.5 | 2.4 | 1.4 | -14 | -9.6 | 1.6 | -1.2 | 4.5 | 8.3 | -87 |
| SOE1+2 | -12.7 | 7.3 | -12.8 | 5.0 | -0.1 | 1 | -14.5 | 6.5 | -7.5 | 4.0 | 7.0 | -48 |
| SOE1a | -18.3 | 1.2 | -15.3 | 3.9 | 2.9 | -16 | -16.1 | 3.6 | -4.4 | 3.6 | 11.8 | -73 |
| SOE3 | | | -7.4 | 3.7 | | | | | -5.7 | 2.9 | | |
| SOE3a | -12.2 | 0.7 | -12.6 | 1.9 | -0.3 | 3 | -10.1 | 1.1 | -3.5 | 2.5 | 6.6 | -65 |
| SOE4 | | | -6.4 | 3.9 | | | | | -5.8 | 3.9 | | |
| PI1+2 | -10.1 | 1.6 | -10.0 | 2.0 | 0.1 | -1 | -9.6 | 1.6 | -7.1 | 2.8 | 2.5 | -26 |
| PI3 | | | -10.6 | 1.8 | | | | | -11.0 | 2.4 | | |
| PI4 | | | -7.6 | 1.9 | | | | | -7.3 | 4.0 | | |
| PI5 | | | -14.7 | 3.1 | | | | | -16.3 | 3.5 | | |
| KT1+2 | -8.6 | 4.1 | -0.5 | 5.5 | 8.2 | -95 | -16.9 | 3.5 | -15.0 | 3.3 | 1.9 | -11 |
| KT3 | | | -3.7 | 2.8 | | | | | -13.9 | 6.5 | | |
| KT4 | | | -1.6 | 3.2 | | | | | -6.2 | 3.2 | | |
| KT5 | -4.7 | 1.6 | -4.0 | 2.0 | 0.7 | -15 | -12.4 | 1.0 | -3.9 | 3.4 | 8.5 | -68 |



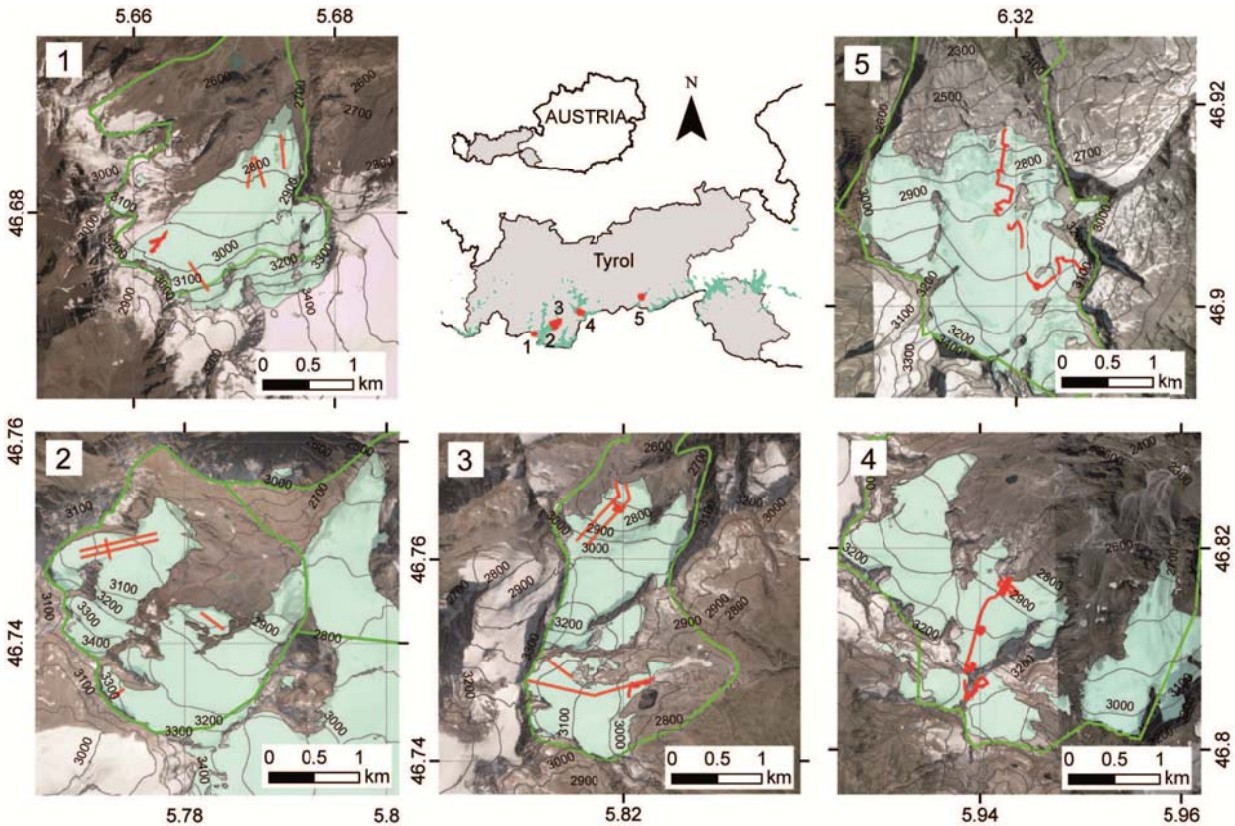

Figure 1. Overview of the Tyrolean glacier ski resorts of Kaunertal (1), Pitztal (2), Sölden (3), Stubai (4) and Hintertux (5). Measurement locations (red line), ski resort outlines (green line), glaciers assigned to the resort (light blue) and contour lines of the GI3 DEMs are presented on orthophotos (tirol.gv.at).

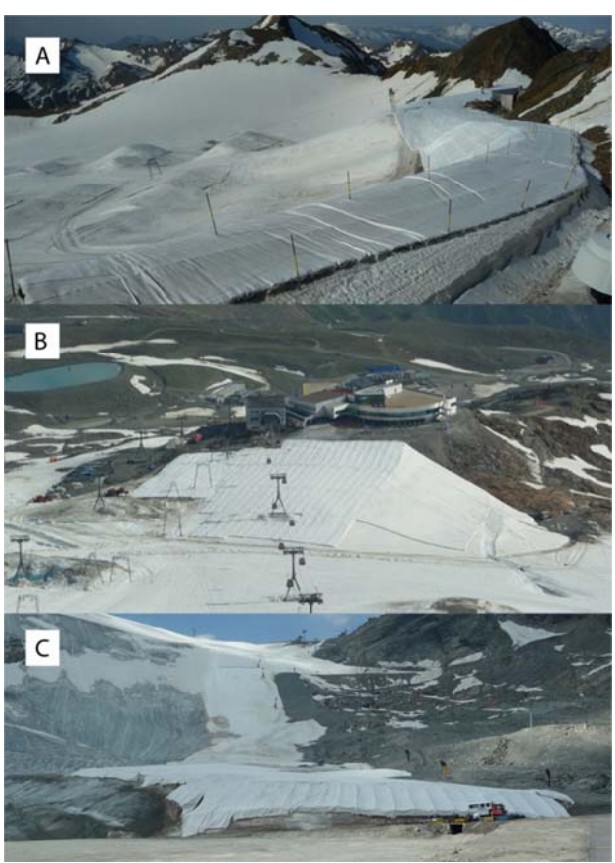

Figure 2. Areas of mass balance management at (A) profile ST1 and (B) profile ST5 in Stubai Glacier ski resort, and at profile SOE2 on Rettenbachferner in Sölden ski resort.





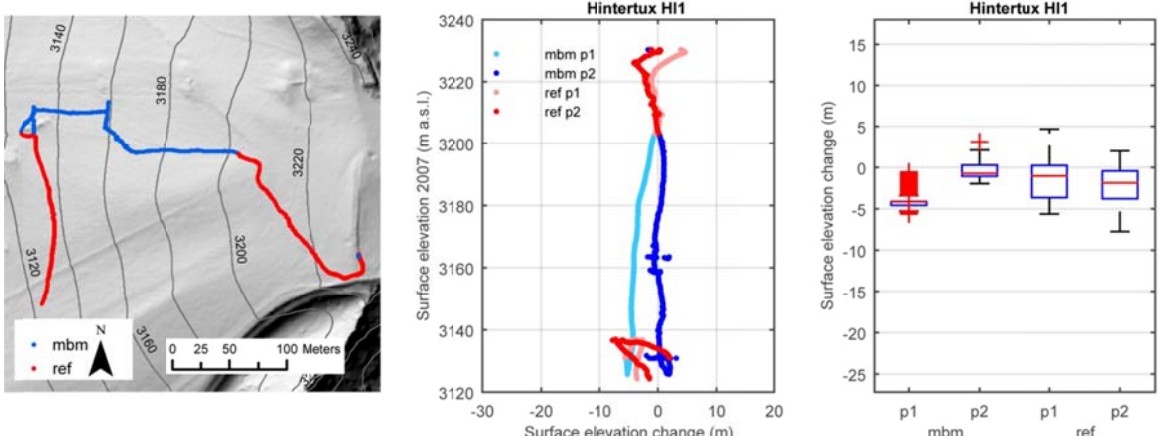

Figure 3. Location of profile HI1 (left), surface elevation changes plotted for surface elevation in 2007 (middle) and boxplot of surface elevation changes along the profile (right, red line: median, blue box: 25/75% percentile, whisker: 1.5 interquartile range, red cross: outliers) separated into area of mass balance management (mbm; in blue) and area without mass balance management (ref; in red) for the periods 1999 -2007 (p1) and 2007 – 2015 (p2).

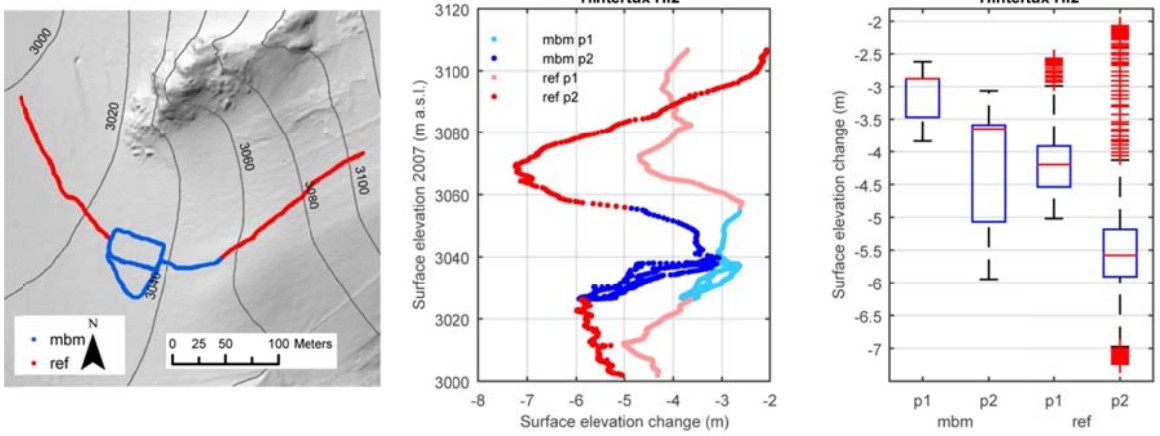

Figure 4. Location of profile HI2, surface elevation changes plotted for surface elevation in 2007 and boxplot of surface elevation changes along the profile separated into area of mass balance management (mbm; in blue) and area without mass balance management (ref; in red) for the periods 1999 -2007 (p1) and 2007 – 2015 (p2).



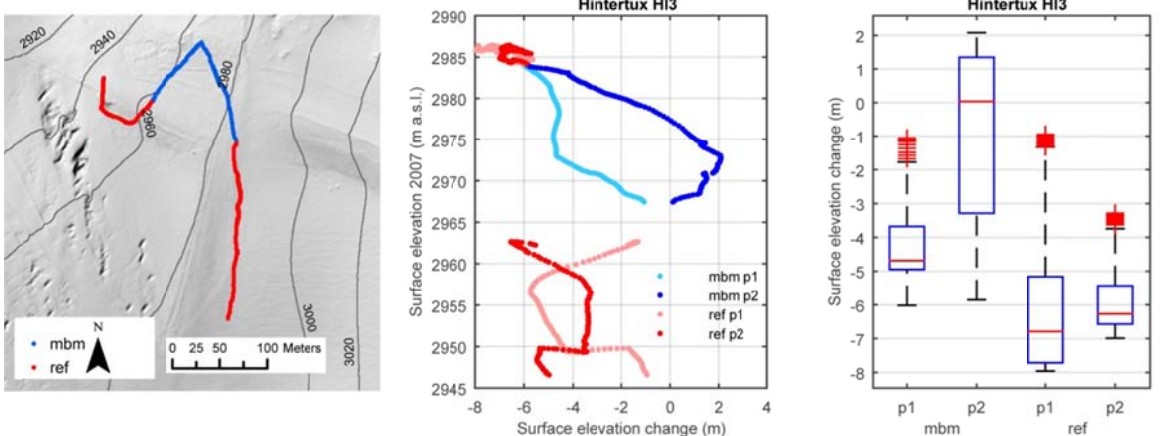

Figure 5. Location of profile HI3, surface elevation changes plotted for surface elevation in 2007 and boxplot of surface elevation changes along the profile separated into area of mass balance management (mbm; in blue) and area without mass balance management (ref; in red) for the periods 1999 -2007 (p1) and 2007 – 2015 (p2).

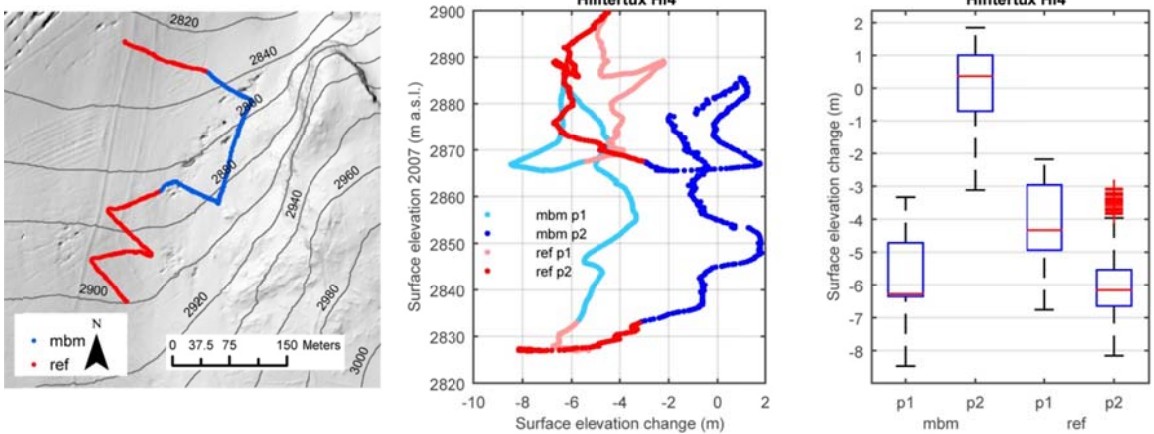

Figure 6. Location of profile HI4, surface elevation changes plotted for surface elevation in 2007 and boxplot of surface elevation changes along the profile separated into area of mass balance management (mbm; in blue) and area without mass balance management (ref; in red) for the periods 1999 -2007 (p1) and 2007 – 2015 (p2).



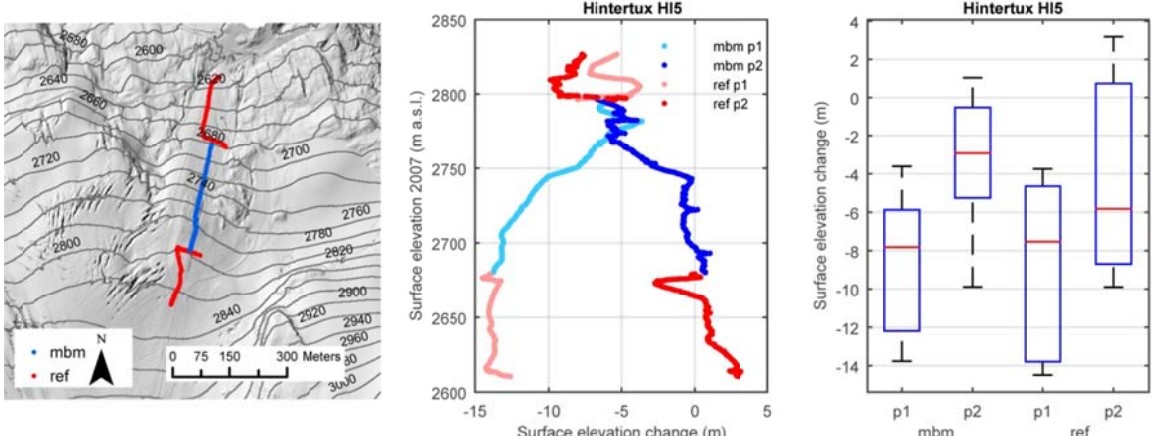

Figure 7. Location of profile HI5, surface elevation changes plotted for surface elevation in 2007 and boxplot of surface elevation changes along the profile separated into area of mass balance management (mbm; in blue) and area without mass balance management (ref; in red) for the periods 1999 -2007 (p1) and 2007 – 2015 (p2).

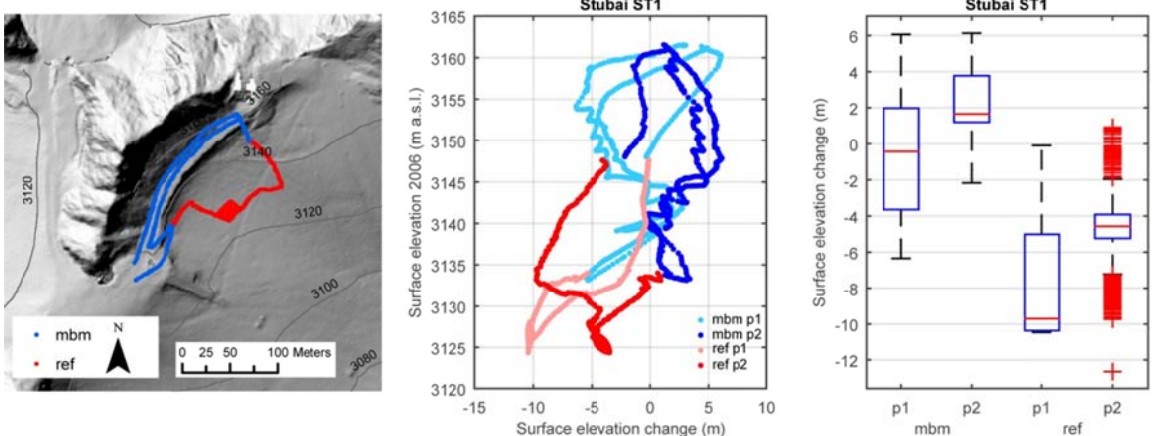

Figure 8. Location of profile ST1, surface elevation changes plotted for surface elevation in 2006 and boxplot of surface elevation changes along the profile separated into area of mass balance management (mbm; in blue) and area without mass balance management (ref; in red)

10   for the periods 1997 -2006 (p1) and 2006 – 2015 (p2).



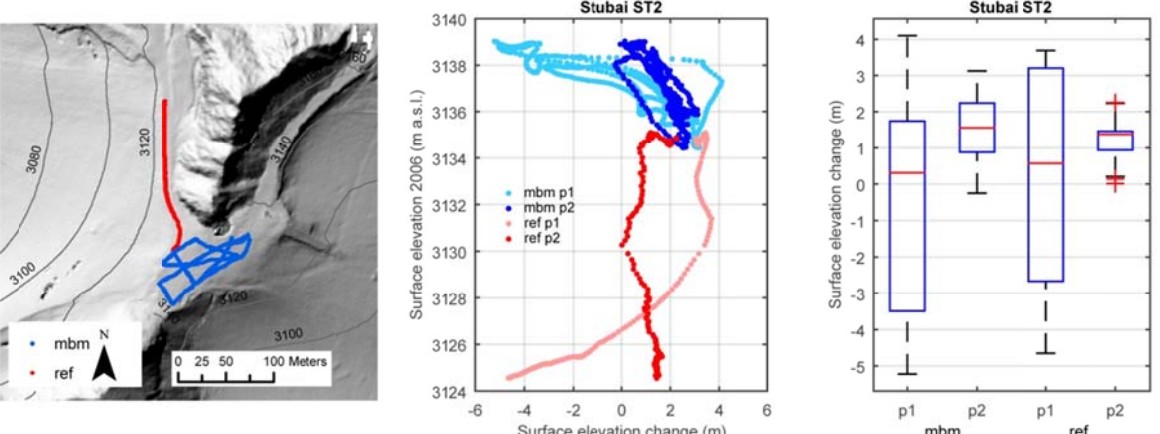

Figure 9. Location of profile ST2, surface elevation changes plotted for surface elevation in 2006 and boxplot of surface elevation changes along the profile separated into area of mass balance management (mbm; in blue) and area without mass balance management (ref; in red) for the periods 1997 -2006 (p1) and 2006 – 2015 (p2).

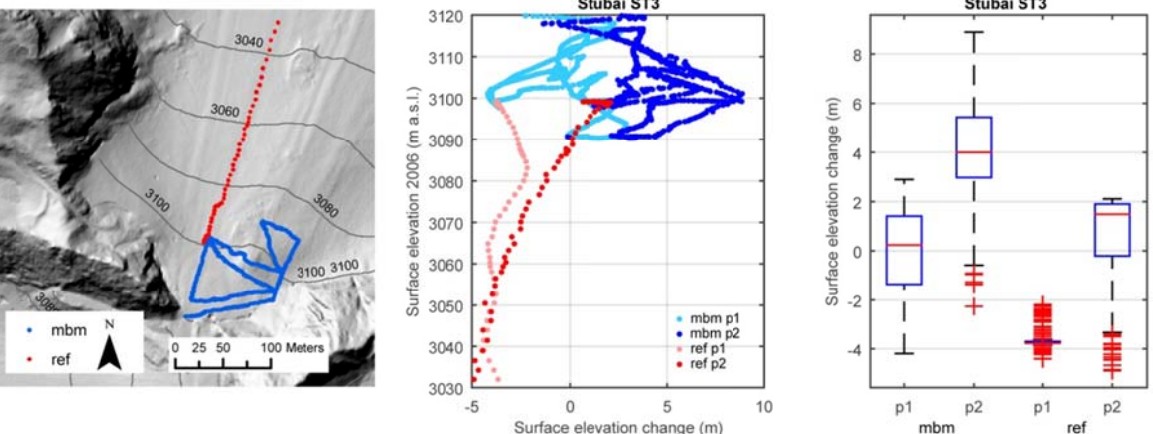

Figure 10. Location of profile ST3, surface elevation changes plotted for surface elevation in 2006 and boxplot of surface elevation changes along the profile separated into area of mass balance management (mbm; in blue) and area without mass balance management

10 (ref; in red) for the periods 1997 -2006 (p1) and 2006 – 2015 (p2).



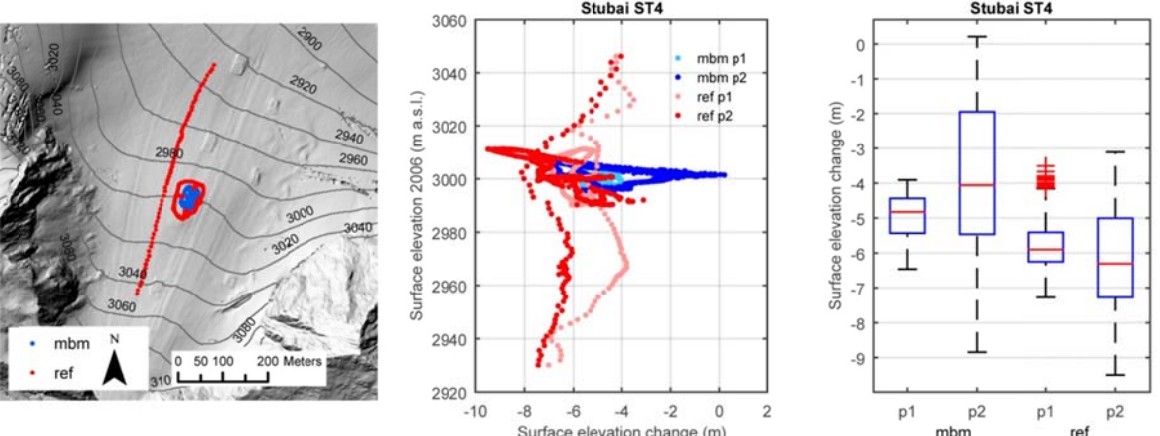

Figure 11. Location of profile ST4, surface elevation changes plotted for surface elevation in 2006 and boxplot of surface elevation changes along the profile separated into area of mass balance management (mbm; in blue) and area without mass balance management (ref; in red) for the periods 1997 -2006 (p1) and 2006 – 2015 (p2).

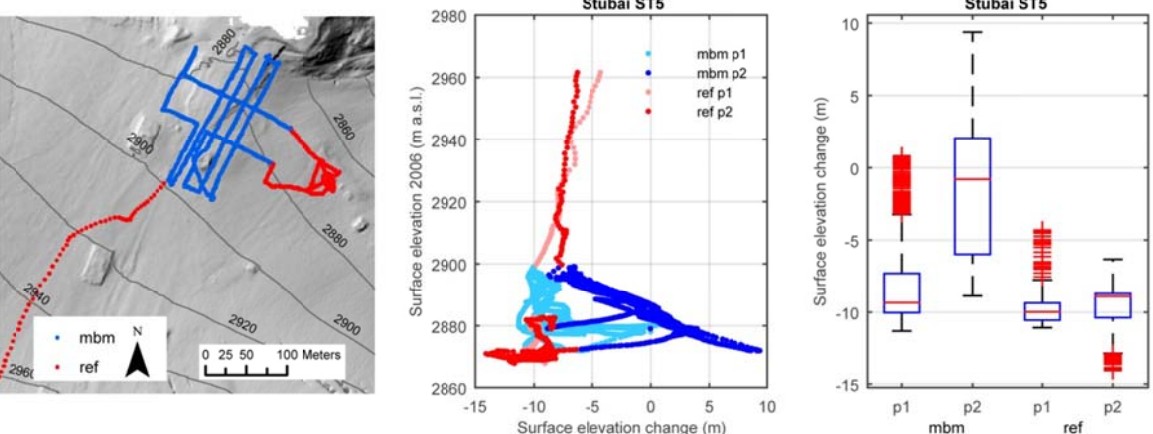

Figure 12. Location of profile ST5, surface elevation changes plotted for surface elevation in 2006 and boxplot of surface elevation changes along the profile separated into area of mass balance management (mbm; in blue) and area without mass balance management

10    (ref; in red) for the periods 1997 -2006 (p1) and 2006 – 2015 (p2).



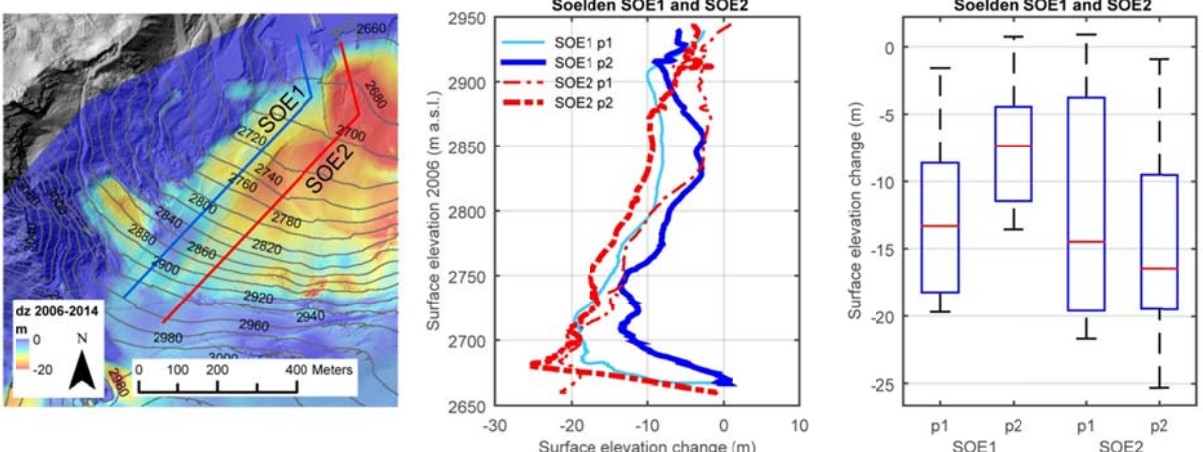

Figure 13. Location of the profiles SOE1 and SOE2 with spatial distribution of surface elevation changes between 2006 and 2014, surface elevation changes plotted for surface elevation in 2006 and boxplot of surface elevation changes along the profile separated into area of mass balance management (mbm; in blue) and area without mass balance management (ref; in red) for the periods 1997 - 2006 (p1) and 2006 – 2014 (p2).

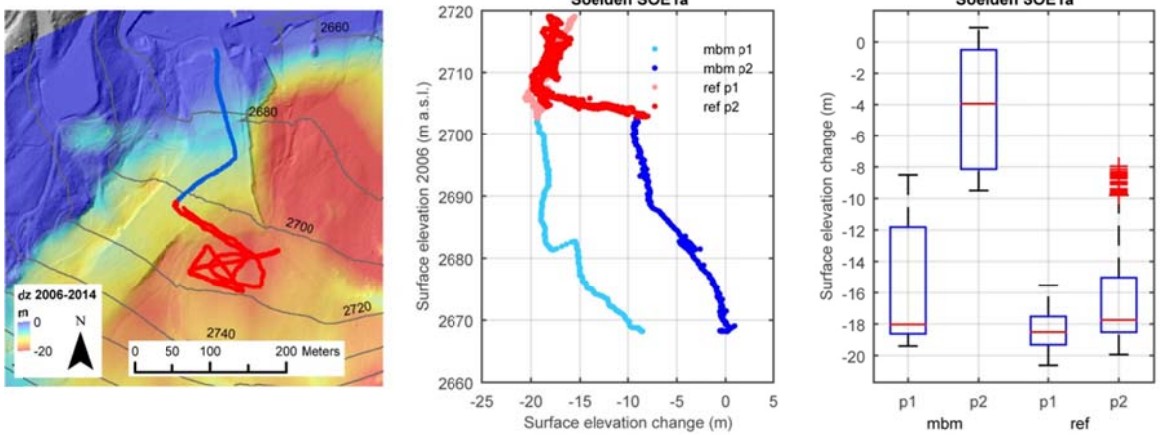

Figure 14. Location of the profile SOE1a with spatial distribution of surface elevation changes between 2006 and 2014, surface elevation changes plotted for surface elevation in 2006 and boxplot of surface elevation changes along the profile separated into area of mass balance management (mbm; in blue) and area without mass balance management (ref; in red) for the periods 1997 - 2006 (p1) and 2006 – 2015 (p2).





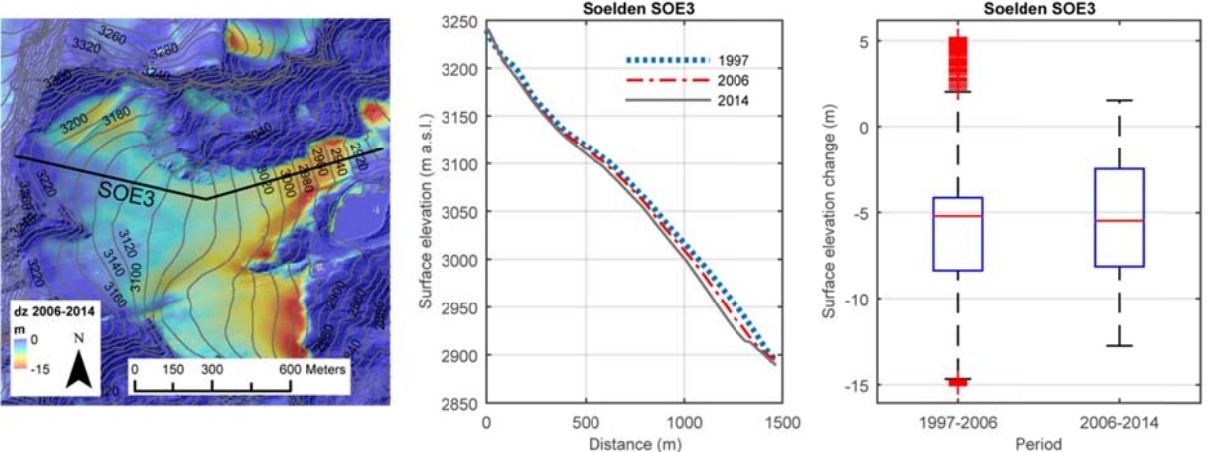

Figure 15. Location of the profile SOE3 with spatial distribution of surface elevation changes between 2006 and 2014, surface elevation along the profile and boxplot of surface elevation changes of the periods 1997 - 2006 and 2006 – 2014.

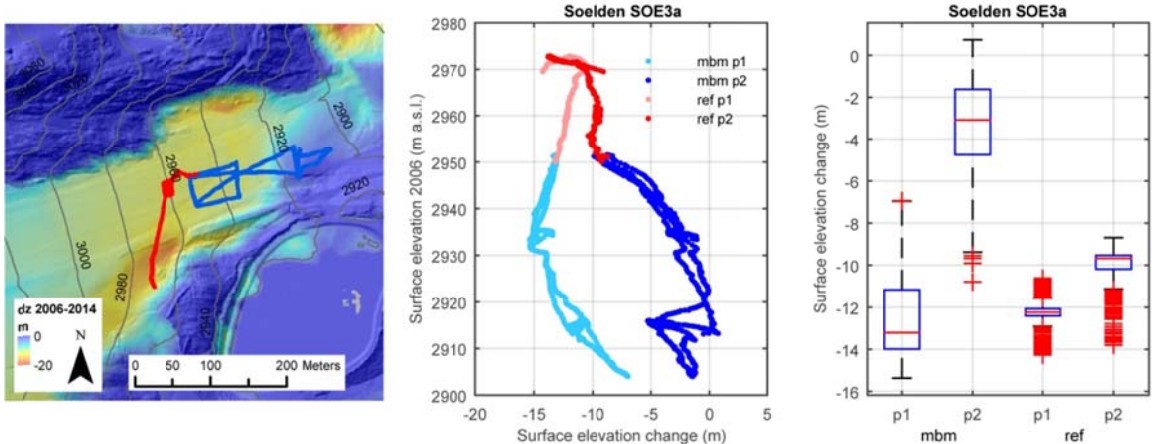

Figure 16. Location of the profile SOE3a with spatial distribution of surface elevation changes between 2006 and 2014, surface elevation changes plotted for surface elevation in 2006 and boxplot of surface elevation changes along the profile separated into area of mass balance management (mbm; in blue) and area without mass balance management (ref; in red) for the periods 1997 - 2006 (p1) and 2006 – 10 2015 (p2).





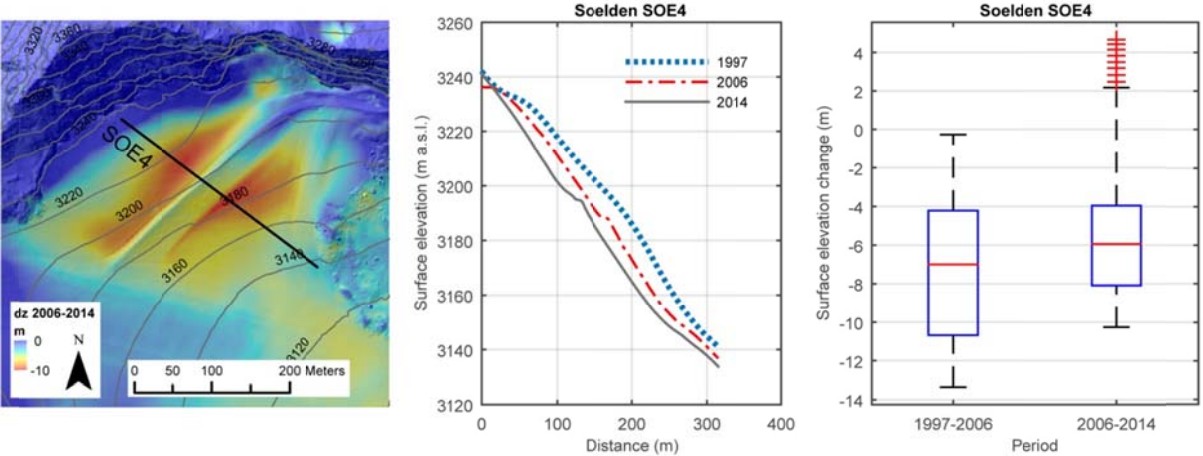

Figure 17. Location of the profile SOE4 with spatial distribution of surface elevation changes between 2006 and 2014, surface elevation along the profile and boxplot of surface elevation changes of the periods 1997 - 2006 and 2006 – 2014.

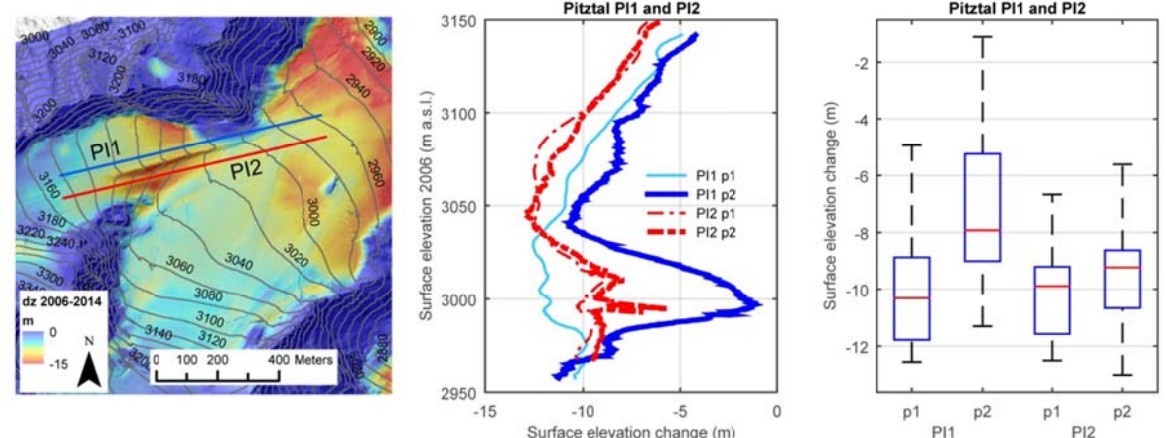

Figure 18. Location of the profiles PI1 and PI2 with spatial distribution of surface elevation changes between 2006 and 2014, surface elevation changes plotted for surface elevation in 2006 and boxplot of surface elevation changes along the profile separated into area of mass balance management (mbm; in blue) and area without mass balance management (ref; in red) for the periods 1997 - 2006 (p1) and
10    2006 – 2014 (p2).





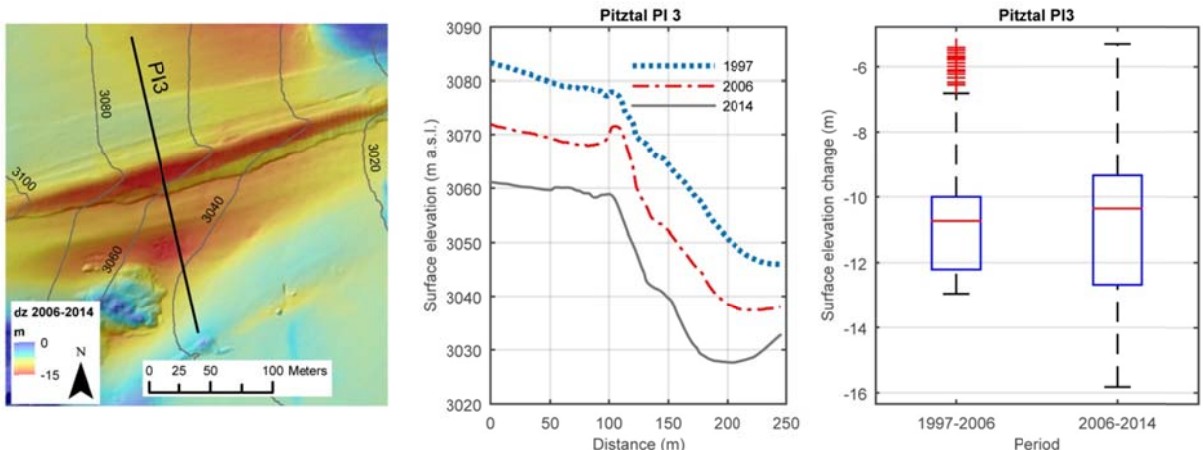

Figure 19. Location of the profile PI3 with spatial distribution of surface elevation changes between 2006 and 2014, surface elevation along the profile and boxplot of surface elevation changes of the periods 1997 - 2006 and 2006 – 2014.

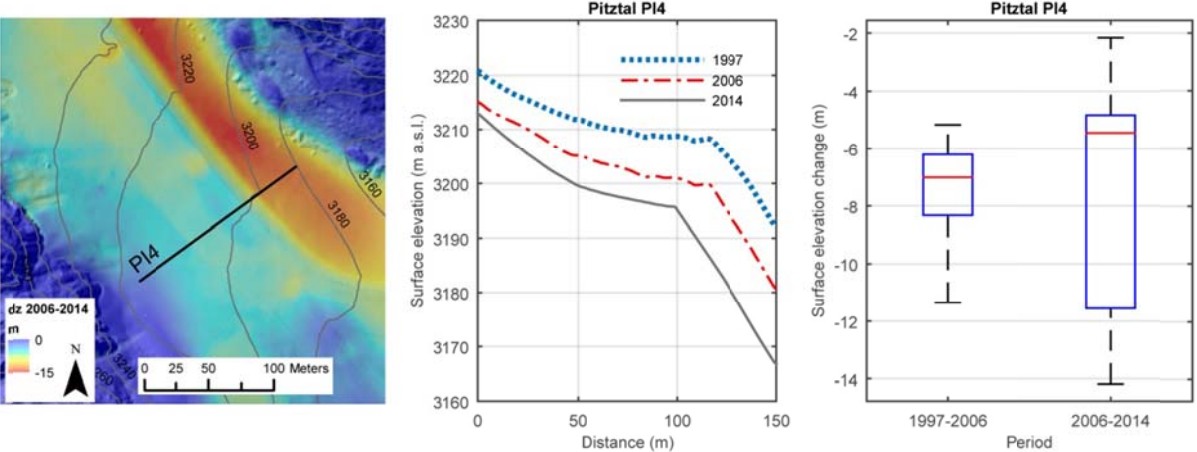

Figure 20. Location of the profile PI4 with spatial distribution of surface elevation changes between 2006 and 2014, surface elevation along the profile and boxplot of surface elevation changes of the periods 1997 - 2006 and 2006 – 2014.



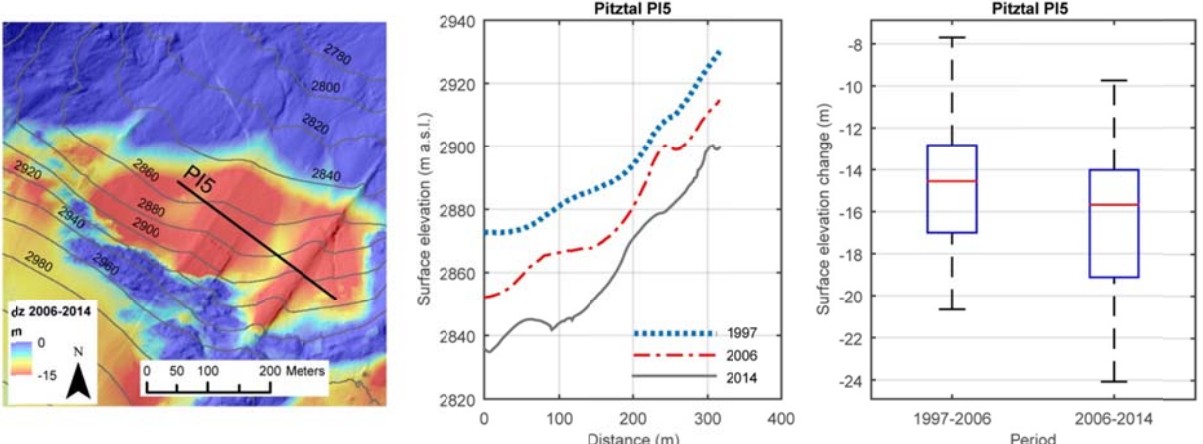

Figure 21. Location of the profile PI5 with spatial distribution of surface elevation changes between 2006 and 2014, surface elevation along the profile and boxplot of surface elevation changes of the periods 1997 - 2006 and 2006 – 2014.

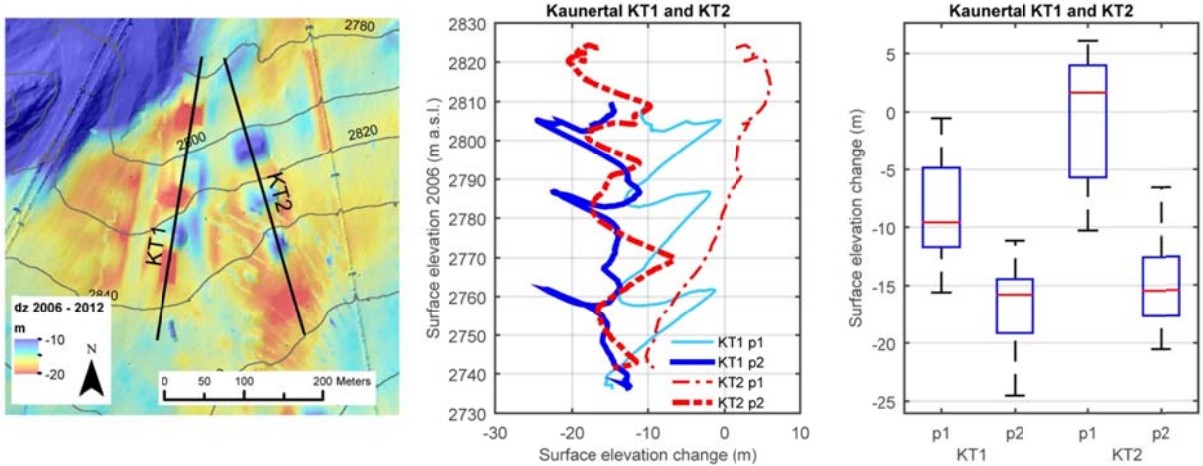

Figure 22. Location of the profiles KT11 and KT2 with spatial distribution of surface elevation changes between 2006 and 2012, surface elevation changes plotted for surface elevation in 2006 and boxplot of surface elevation changes along the profile separated into area of mass balance management (mbm; in blue) and area without mass balance management (ref; in red) for the periods 1997 - 2006 (p1) and
10    2006 – 2012 (p2).



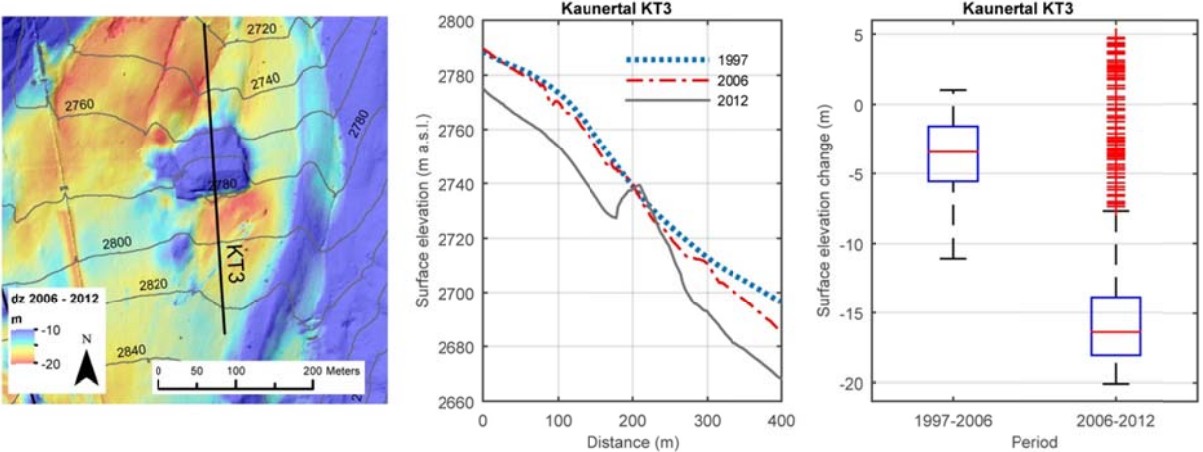

Figure 23. Location of the profile KT3 with spatial distribution of surface elevation changes between 2006 and 2012, surface elevation along the profile and boxplot of surface elevation changes of the periods 1997 - 2006 and 2006 – 2012.

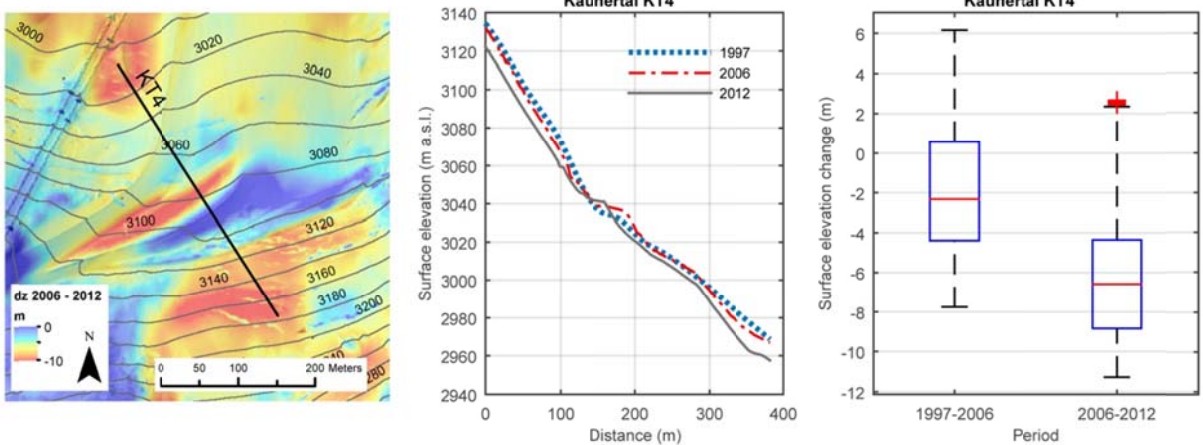

Figure 24. Location of the profile KT4 with spatial distribution of surface elevation changes between 2006 and 2012, surface elevation along the profile and boxplot of surface elevation changes of the periods 1997 - 2006 and 2006 – 2012.





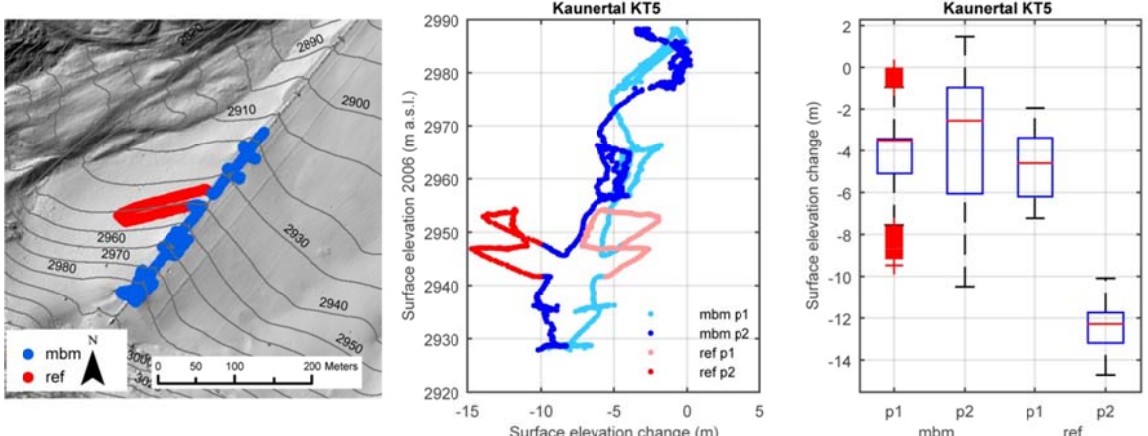

Figure 25. Location of profile KT5, surface elevation changes plotted for surface elevation in 2006 and boxplot of surface elevation changes along the profile separated into area of mass balance management (mbm; in blue) and area without mass balance management (ref; in red) for the periods 1997 -2006 (p1) and 2006 – 2015 (p2).

