# Peer review of "Local reduction of decadal glacier thickness loss through mass balance management in ski resorts"

_The Cryosphere, 2016_

## Referee Comment (RC1) · Anonymous Referee #1 · 27 May 2016

General comments :

This paper deals with the impact of mass balance management on glacier thickness changes in ski resorts. From photogrammetry, laser scanning and GPS measurements, the authors compared the thickness changes on profiles with and without mass balance measurements over the last 20 years. The authors conclude that thickness changes could be reduced by 35-65% thanks to the mass balance management. This paper shows a large dataset given that 16 profiles on 5 glaciers have been measured since 1997 or 1999. These comparisons are rare on the alpine glaciers and these results certainly deserve to be published. It does not concern the scientific community only but also many people involved in the mass balance management.

[Figure]

However, this manuscript has large weaknesses and did not reach a sufficient maturity. This manuscript is difficult to read and confusing.

First, the authors should revise the structure: - Data: a lot of information should be included in Data and not elsewhere in the paper: for instance, the information related to the uncertainties on photogrammetry, GPS.., measurements given in Discussion (lines 5-15 p 10) should be reported in Data section. The authors should check that, everywhere in the manuscript. Seven lines in "Surface elevation data" are not sufficient to describe the measurements given these data are the basis of the paper. The authors should explain here clearly that DGPS measurements of 2014/2015 are compared to DEM from 1997/1999 and 2006/2007. It is not obvious at this stage of the mansucript. - The techniques of management on each glaciers should be summarized in a Table (maybe in the Table 2). - In Data and Methods section, the explanations about the emergence velocities (p.5, lines 1-20) should be removed from Data and Methods: first, the authors do not provide any explanations here why and how they used these equations. At this stage, the reader wonders why the authors introduce these Equations relative to the emergence/submergence velocities. These equations should be moved to the Discussion (lines 16-29, p10) where the authors provide a discussion about the relationship between the surface mass balance and the elevation changes. However, I am not sure these Equations are helpful given the authors do not use them. In any case, the authors should use the classical way to present the equation related to emergence velocity (Cuffey and Paterson, 2010, equation 8.65). Equations 2 and 3 are not necessary in any case, given these equations are not used for calculations in this paper. - Study sites: the authors should replace the long (and indigestible) description by a Table. - Results: this section is indigestible. The reader does not need the full and detailed description of elevation changes at each pylons, skilifts, pistes.... Here, the description seems to come directly from a technical report. It is not useful for the scientific community. The number of Figures which show the elevation changes (Fig 3, 4, 56,7,8, 9, 10, 11, 12, 13, 14, 15, 16, 17, 18, 19, 20, 21, 22, 23,24 and 25 !!) should be considerably reduced and most of them should be moved in a supplementary material. For this section, the authors should make a strong effort to sum up the results, to analyze them and to make a new Figure to show the summarized results. From my point of view, it is absolutely necessary and this kind of Figure would be useful for the scientific community.

Second, the analysis of results is poor. I am aware of the difficulties given that the data come from different techniques and different areas. In this way, it is very difficult to compare elevation changes for areas with different altitudes and different aspect. However, despite on Table 3, there is a lack of quantitative results. I think that Table 3 is not sufficient to analyze the results. In addition, I am not sure that results given in relative reduction % are relevant. Some figures in Table 3 seem to me strange or wrong: for instance, at ST2, the authors reported a relative reduction of -396% despite on the fact that the elevation change is +0.2 m for the reference profile and -0.7 m for the profile with mass balance management. Did I miss something? If not, the authors should check the whole results. Again, I do not think the relative reduction in % is meaningfull. When the value of elevation changes are close to zero (close to ELA), the relative reduction can reach very large values but it does not mean that the impact is more important. This way of presenting the results is not convincing. I believe that the percentages given in the manuscript (and in Abstract) are easy to understand for the general public but are probably no relevant.

Third, I am not convinced by the conclusions relative to the impacts of mbm. For instance, the authors claimed that "the submergence and emergence should be similar. . .so that a large impact resulting from different or changing ice flow regimes is unlikely". It can be questioned from the results shown in this study. For instance, it seems very difficult to make conclusions about the impact of mass balance management when the measurements have been done at very different altitudes (Fig. 3, Fig 5, Fig. 7. . ..) and for different aspects. Moreover, I do not understand how the submergence/emergence velocities spatial distribution can be neglected in this study. The authors wrote that "Interannual differences in emergence/submergence velocities are

less than 0.5 m a-1 at Kesselwandferner", but, here, this is the spatial distribution of emergence velocities which is questioned. Or I missed something. The results shown in Fig 3 to 25 are confusing and again, a thorough analysis and a synthesis are missing to provide relevant results and to convince the reader.

Detailed comments:

Many things should be improved but I think it is not necessary to make a list at this stage given that the structure of the manuscript and the analysis of the results should be strongly revised first. Detail information should be removed from the manuscript when there are not used in the manuscript (GPR data, history of ski tourism...). The authors should check that carefully.

---

## Short Comment (SC1) · 3 Jun 2016

Fischer et al (2016) provide by far the most extensive examination of the impact of ski area management on local glacier mass balance. This is a unique data set that cannot be matched elsewhere; hence this contribution provides a valuable snapshot at a critical moment for ski areas with glacier terrain response to climate change. Most of the comments below are quite minor. Considerable figure consolidation could be completed. Brief reference to the practice in other nations is warranted. Also the impact of new snow and grooming on increasing albedo should be mentioned, even though, the point of this study was not to quantify that impact.

2-1: to store and maintain snow. . ..

[Figure]

2-13: The to They

2-17: Not only has visitor demand developed over time but cable car technology has advanced. . .

3-3: Crevassed reduced not just at ski areas but on other glaciers too, for example Colgan et al (2016) Pelto and Hedlund (2001)

3-5: Is removal of rock, sand and dirt from the piste not a goal? Grooming and new snow production oth increase the albedo. This is a goal noted by some of your previous research.

7-26: I assume the 35% and 65% reduction are compared to adjacent areas of the same glaciers, if so more clearly state this. Somewhere it would helpful to reference typical thickness loss values from either WGMS reporting Austrian glaciers or from the inventory, as a wider reference.

9-2: Continuous grooming will increase albedo.

10-26: I agree with this assertion "In any case, submergence and emergence should be similar for the profiles and the reference profiles"

11-2: Grooming would also reduce albedo.

12-3: It is worth noting that mass balance management extends to Tignes, France; Whistler, BC and Mount Hood, OR.

12-28: The enhanced prominence of managed area versus managed areas, generates steeper slopes as noted. This in turn should increase ablation. Will also act both as a wind scour and potentially wind trap for accumulation. Is this observed?

Figure 1: Ski area boundary line should be more distinct color.

Figures: The number of profile figures is impressive. However, collectively they are redundant and also detract from highlighting important overall trends. The variation

from profile to profile becomes the focus. I would suggest utilizing only two sets from each glacier, or focusing more on the central panel. The central panel alternative takes advantage of the fact that Table 3 provides the data from the third panel for each glacier. Figure 1 provides profile location. Hence, you could just use the middle panel for all but two profiles on each glacier.

Colgan, W., H. Rajaram, H., Abdalati, W., McCutchan, C., Mottram, R., Moussavi, M. and Grigsby, S: Glacier crevasses: Observations, models, and mass balance implications, Rev. Geophys., 54, 119–161, doi:10.1002/2015RG000504, 2016.

Pelto, M.S., and Hedlund, C.: The terminus behavior and response time of North Cascade glaciers. Journal of Glaciology 47: 497–506, 2001.
* * *

---

## Referee Comment (RC2) · Anonymous Referee #2 · 7 Jun 2016

General Comments

The paper presents a valuable, comprehensive and comprehensible overview about the medium-term (decadal) effect of technical modifications of the glacier surface mass balance within Austrian Skiing resorts. The application of these measures started around the year 2004 and the related physical processes and short-term effects were already investigated in detail in a number of earlier studies. The authors analyze digital elevation model differences as well as DGPS measurements at selected spots of different glaciers with and without application of such measures between multiple years in order to quantify the effect of these intentional modifications on surface elevation changes within this timescale. Results indicate the clear medium-term benefit as well

as the limitations of these technical measures on a larger scale in terms of costs and efforts. Although the uncertainty of their method is discussed in the manuscript, the latter should be done in a more thorough, quantitative way, thereby also using an appropriate and exact terminology. In a revised version of the manuscript, the individual uncertainty sources should not only be named but all of them also be estimated and the resultant combined expanded uncertainty as well as its impact on the main results of the paper calculated. Therefore I suggest accepting the paper after the points listed in the specific comments and some minor ones in the technical corrections have been implemented by the authors.

Specific Comments (in decreasing order of importance)

(1) In the discussion section (p10. Lines 5-15) the Authors indicate a maximum uncertainty for their method of 1.1 m for both the DGPS and the DEM differences. It is not clear a) how this number is calculated exactly (uncertainty components), b) what confidence interval it is referred to (e.g. standard (66%) or expanded (95% level) uncertainty), c) what the impact of the combined expanded uncertainty is on the main results of the paper. For clarity and consistency, I very much encourage the Authors to study and use the Guide to the Expression of Uncertainty in Measurement (GUM; JCGM, 2008)) as well as the terminology that is defined therein.

(2) It is not clear how areas with long-term mass balance management were exactly identified (onsite location) in the study (own (GPS) records or data from skiing resorts?,...). Please add this information.

(3) Concerning the single effect of grooming on snow and ice ablation, the authors should add that the observed effect was in the order of only 5 % rather than 10% and that this number was very close to the measurement uncertainty (Olefs and Fischer, 2008; Fischer et al., 2011; ;Olefs, 2005;Olefs and Obleitner, 2007). It is also worth to clarify the following in the paper: Based on previous studies, it is still not clear what exact physical mechanism(s) leads to the observed effect. Beside the reduction of

surface layer erodibility through compaction (stronger bonding of the snow crystals), there may be other effects, e.g. a modification of surface albedo due to a reduction of average grain size of the surface snow layers induced by the snow-cat or a modification of snow thermal conductivity (Olefs and Obleitner, 2007). If there are new studies that separate those exact effects on the ablation reduction known to the authors, they should cite them.

(4) I strongly suggest adding units (SI) to all variables whenever formulas or variables are used in the manuscript (e.g. p.5).

(5) The physical effect of water injection in the snow cover is mainly to add mass to the existing seasonal snow (if there is enough cold content in the snow to refreeze the injected water). After injection, the release of latent heat due to refreezing of the water decreases the absolute value of the cold content of the existing snow cover (as e.g. shown in Fig.7 of Olefs and Fischer, 2008). Firstly, I do not understand why the cold content should be increased by this method (p.3, line 19). Secondly, the authors could also add the two main resulting limitations of this method apart from the large effort: enough cold content before injection and timing problem (enough time between applications).

Technical corrections

p1 (1) l30: "...depth height...??" (2) l4: Fischer et al., 2011 a or b? p2 (3) l6: please explain the first occurrence of the shortcut "GI" (4) l9: 1987 in the manuscript, 1986 in Tab.1 ?? (5) l13: They noticed... (6) l18 and others: I would prefer "t-bar lift" instead of "tow lift" throughout the paper p3 (7) l19: increase or decrease cold content? you could use the absolute value to clarify... p4 (8) l19: please use consistent naming for "Austrian glacier inventory" (GI?) p5 (9) please add units to all variables (SI) p6 (10) For DGPS (?) profiles... P7 (11) L20: here the single effect of grooming (compaction of the surface layers) is mixed with the potential effect of snow farming (lateral transport of snow mass by snow cats), please clarify. (12) L27: (Tab.3) P8 (13) L29-30: "On

average" instead of "In mean" P12 (14) l5:...ski tourism in the year 2100... (15) L12: "Fujita and Ageta, 2000" is missing in the references (16) L16: I would suggest to write "(e.g. without glacier cover)" as a) in other regions of the world glaciers do exist at low altitudes and b) the fact that the effectiveness of surface textiles to reduce ablation decreases with altitude is not tied to the surface type (glacier or not) but it is due to the energy balance being dominated by sensible/latent heat fluxes at lower altitudes. (17) L18: at the end of this sentence you could again cite the work of Skogsberg as well as Grünewald and Wolfsperger).

References

P15, l21: The year of publication should be placed at the end.

Figures

Fig.1: In the caption please specify whether DGPS measurements are indicated by the red lines. Fig.3 and following: it is not clear what you mean with "surface elevation changes plotted for surface elevation in 2007" ? Do you mean the difference 2007 – 1999 and 2015-2007 ? Please clarify in the captions and also in the ordinate label. Fig3.: It shoud be 25th /75th percentile (and not %!)

The captions of all following figures could be reduced...there is a lot of redundant information. Fig.13 and 18: on the right subplot "mbm" and "ref" is missing as label

References

Joint Committee for Guides in Measurements (JCGM): Evaluation of measurement data – Guide to expression of uncertainty in measure-ment, JCGM 100:2008, GUM 1995 with minor corrections, available at: http://www.bipm.org/utils/common/documents/jcgm/JCGM_100_2008_E.pdf, 2008.

---

## Editor Comment (EC1) · C. R. Stokes (Editor) · 21 Jun 2016

I would like to thank those who have given up their time to provide some very helpful comments on this manuscript. There is consensus that this manuscript represents an important contribution, but that it requires some substantial revisions. Thus, I would certainly welcome a revised manuscript, but would likely send it back out for re-review.

---

## Author Comment (AC2) · 14 Jul 2016

*Reviewer comments of M. Pelto mauri.pelto@nichols.edu are italic and underlined*

responses

*Fischer et al (2016) provide by far the most extensive examination of the impact of ski area management on local glacier mass balance. This is a unique data set that cannot be matched elsewhere; hence this contribution provides a valuable snapshot at a critical moment for ski areas with glacier terrain response to climate change. Most of the comments below are quite minor. Considerable figure consolidation could be completed. Brief reference to the practice in other nations is warranted. Also the impact of new snow and grooming on increasing albedo should be mentioned, even though, the point of this study was not to quantify that impact.*

The figures now contain overview graphs and just few examples of the different data sets. The original Figures are shifted to the supplement still as a part of the draft.

We cited all the papers we could find on practice in glacier ski resorts in other nations, and would really like to add additional literature to this topic. Is there citable literature describing the mass balance management on the sites you mention?

We added the following citation to describe the effect of grooming on albedo:

Keller, T. Pielmeier, C. Rixen, C. Gadient, F. Gustafsson, D. Stähli, M., 2004. Impact of artificial snow and ski-slope grooming on snowpack properties and soil thermal regime in a sub-alpine ski area. Annals of Glaciology, 38, 1, 314-318.

Unfortunately, Keller et al. do not find empirical data on the effect of grooming on snow albedo. Measurements of this effect might be difficult for various reasons, amongst them:

- Bidirectional reflection of snow requires careful consideration of all combinations of grooming tracks and incidence angles
- During operation, ski tracks replace grooming tracks, with even more difficult to capture changes in optical properties.

[Figure]

Figure 1: A single skier changes surface albedo on 26.05.2005 in 2850 m.

Taking into account, that grooming during the investigated period ended with mid to end of May, and surface melt processes had been observed mainly from beginning of May onwards, the effect of grooming on albedo during melting season is thus small, which was the reason for skipping that topic initially.

*2-1: to store and maintain snow:::.*

changed

*2-13: The to They*

changed

*2-17: Not only has visitor demand developed over time but cable car technology has advanced::.*

changed

*3-3: Crevassed reduced not just at ski areas but on other glaciers too, for example Colgan et al (2016) Pelto and Hedlund (2001).*

We agree, this is stated also for example in Fischer, A. (2010) Glaciers and climate change: Interpretation of 50 years of direct mass balance of Hintereisferner, Global and Planetary Change 71, 1-2: 13-26.

*3-5: Is removal of rock, sand and dirt from the piste not a goal? Grooming and new snow production oth increase the albedo. This is a goal noted by some of your previous research.*

These measures have been included in the first point which comprises measures to decrease surface roughness. We added explicit examples to that point. The albedo topic is included in the third point, reduction of mass loss, as the high albedo is not an aim on its own purpose, but for its influence on mass balance. But as stated before, grooming during ablation season is rare.

*7-26: I assume the 35% and 65% reduction are compared to adjacent areas of the same glaciers, if so more clearly state this. Somewhere it would helpful to reference typical thickness loss values from either WGMS reporting Austrian glaciers or from the inventory, as a wider reference.*

We restructured the article including a better description of the reference measurements. Typical losses from the glacier inventory have been added.

Abermann, J., Lambrecht, A., Fischer, A., and Kuhn, M.: Quantifying changes and trends in glacier area and volume in the Austrian Ötztal Alps (1969-1997-2006), The Cryosphere, 3, 205-215, doi:10.5194/tc-3-205-2009, 2009.

The thickness of Ötztal glaciers reduced by 0.95 m/year in average between 1969 and 1997, and -0.91 m/year between 1997 and 2006.

*9-2: Continuous grooming will increase albedo.*

This section was restructured. The albedo discussion was included in the state of the art report. Unfortunately we do not have data on surface albedo.

*10-26: I agree with this assertion "In any case, submergence and emergence should be similar for the profiles and the reference profiles"*

We rephrased this part.

*11-2: Grooming would also reduce albedo.*

We added the information that grooming takes place during winter, and the effect is measured in summer, so that a direct influence of grooming on albedo is not very likely.

*12-3: It is worth noting that mass balance management extends to Tignes, France; Whistler, BC and Mount Hood, OR.*

We added this information together with other ski resorts from our personal knowledge.

*12-28: The enhanced prominence of managed area versus managed areas, generates steeper slopes as noted. This in turn should increase ablation. Will also act both as a wind scour and potentially wind trap for accumulation. Is this observed?*

Yes, both effects are observed, but not quantified in detail. The removal of snow from previously covered platforms leads to their rapid meltdown once the maintenance stops. The separation of radiative and wind drift effects is hard to measure and drawing general conclusions included high uncertainties.

*Figure 1: Ski area boundary line should be more distinct color.*

changed

*Figures: The number of profile figures is impressive. However, collectively they are redundant and also detract from highlighting important overall trends. The variation from profile to profile becomes the focus. I would suggest utilizing only two sets from each glacier, or focusing more on the central panel. The central panel alternative takes TCD advantage of the fact that Table 3 provides the data from the third panel for each glacier.*

We restructured the Figures.

*Figure 1 provides profile location. Hence, you could just use the middle panel for all but two profiles on each glacier.*

Figure 1 is changed, and we hope that you find it benfical.

*Colgan, W., H. Rajaram, H., Abdalati, W., McCutchan, C., Mottram, R., Moussavi, M. and Grigsby, S: Glacier crevasses: Observations, models, and mass balance implications, Rev. Geophys., 54, 119–161, doi:10.1002/2015RG000504, 2016.*

included

*Pelto, M.S., and Hedlund, C.: The terminus behavior and response time of North Cascade glaciers. Journal of Glaciology 47: 497–506, 2001.*

included

---

## Author Comment (AC3) · 14 Jul 2016

Response to the *anonymous referee #2*

General Comments

*The paper presents a valuable, comprehensive and comprehensible overview about the medium-term (decadal) effect of technical modifications of the glacier surface mass balance within Austrian Skiing resorts. The application of these measures started around the year 2004 and the related physical processes and short-term effects were already investigated in detail in a number of earlier studies. The authors analyze digital elevation model differences as well as DGPS measurements at selected spots of different glaciers with and without application of such measures between multiple years in order to quantify the effect of these intentional modifications on surface elevation changes within this timescale. Results indicate the clear medium-term benefit as well as the limitations of these technical measures on a larger scale in terms of costs and efforts.*

*Although the uncertainty of their method is discussed in the manuscript, the latter should be done in a more thorough, quantitative way, thereby also using an appropriate and exact terminology. In a revised version of the manuscript, the individual uncertainty sources should not only be named but all of them also be estimated and Interactive the resultant combined expanded uncertainty as well as its impact on the main results comment of the paper calculated.*

We restructured the manuscript and added thorough calculation of measurement uncertainties as well as a better description of other uncertainties.

*Therefore I suggest accepting the paper after the points listed in the specific comments and some minor ones in the technical corrections have been implemented by the authors.*

*Specific Comments (in decreasing order of importance)*

(1) *In the discussion section (p10. Lines 5-15) the Authors indicate a maximum uncertainty for their method of 1.1 m for both the DGPS and the DEM differences. It is not clear a) how this number is calculated exactly (uncertainty components), b) what confidence interval it is referred to (e.g. standard (66%) or expanded (95% level) uncertainty), c) what the impact of the combined expanded uncertainty is on the main results of the paper. For clarity and consistency, I very much encourage the Authors to study and use the Guide to the Expression of Uncertainty in Measurement (GUM; JCGM, 2008)) as well as the terminology that is defined therein.*

The discussion of errors and uncertainties was shifted to the section on Data and methods and expanded, discussing systematic and measurement errors separately.

*(2) It is not clear how areas with long-term mass balance management were exactly identified (onsite location) in the study (own (GPS) records or data from skiing resorts?). Please add this information.*

The criteria for the selection of the test sites are now part of the paragraph on test sites. The locations have been clearly indicated by previous (own) GPS records and documents of the pre-projects. Of course only own DGPS measurements are presented here.

*(3) Concerning the single effect of grooming on snow and ice ablation, the authors should add that the observed effect was in the order of only 5 % rather than 10% and that this number was very close to the measurement uncertainty (Olefs and Fischer, 2008; Fischer et al., 2011; ;Olefs, 2005;Olefs and Obleitner, 2007).*

Done – we changed the number to the original 6% and added the information that this is close to the measurement accuracy.

*It is also worth to clarify the following in the paper: Based on previous studies, it is still not clear what exact physical mechanism(s) leads to the observed effect. Beside the reduction of surface layer erodibility through compaction (stronger bonding of the snow crystals), there may be other effects, e.g. a modification of surface albedo due to a reduction of average grain size of the surface snow layers induced by the snow-cat or a modification of snow thermal conductivity (Olefs and Obleitner, 2007). If there are new studies that separate those exact effects on the ablation reduction known to the authors, they should cite them.*

We changed the wording to distinguish measurements and unknowns.

*(4) I strongly suggest adding units (SI) to all variables whenever formulas or variables are used in the manuscript (e.g. p.5).*

The formula is replaced.

*(5) The physical effect of water injection in the snow cover is mainly to add mass to the existing seasonal snow (if there is enough cold content in the snow to refreeze the injected water). After injection, the release of latent heat due to refreezing of the water decreases the absolute value of the cold content of the existing snow cover (as e.g. shown in Fig.7 of Olefs and Fischer, 2008). Firstly, I do not understand why the cold content should be increased by this method (p.3, line 19).*

*Secondly, the authors could also add the two main resulting limitations of this method apart from the large effort: enough cold content before injection and timing problem (enough time between applications).*

We agree with that points ( cold content can increase only if small amounts of water are injected and cold air can penetrate the snow cover through the holes for a longer time; no skiing and no grooming and no snow falls after the injection). As injection is not used as mass balance management method, but mainly for ski racing, we decided for shortening that part rather than going to deep into that topic.

*Technical corrections*

*p1 (1) l30: ":::depth height:::??"*

height removed

*(2) l4: Fischer et al., 2011 a or b?*

a, J Glac

*p2 (3) l6: please explain the first occurrence of the shortcut "GI" (4)*

done

*l9: 1987 in the manuscript, 1986 in Tab.1 ??*

1986

*(5) l13: They noticed::.*

Section rephrased

*(6) l18 and others: I would prefer "t-bar lift" instead of "tow lift" throughout the paper*

replaced

*p3 (7) l19: increase or decrease cold content? you could use the absolute value to clarify::.*

Water injection is not discussed any more

*p4 (8) l19: please use consistent naming for "Austrian glacier inventory" (GI?)*

ok

*p5 (9) please add units to all variables (SI)*

The section is removed

*p6 (10) For DGPS (?) profiles::.*

The section is removed

*P7 (11) L20: here the single effect of grooming (compaction of the surface layers) is mixed with the potential effect of snow farming (lateral transport of snow mass by snow cats), please clarify.*

The section was rewritten. In any case, we can not separate the effects of various measures in this spatially distributed study.

*(12) L27: (Tab.3)*

changed

*P8 (13) L29-30: "On average" instead of "In mean"*

changed

*p12 (14) l5::::ski tourism in the year 2100::.*

changed

*(15) L12: "Fujita and Ageta, 2000" is missing in the references*

added

*(16)L16: I would suggest to write "(e.g. without glacier cover)" as a) in other regions of the world glaciers do exist at low altitudes and b) the fact that the effectiveness of surface textiles to reduce ablation decreases with altitude is not tied to the surface type (glacier or not) but it is due to the interactive energy balance being dominated by sensible/latent heat fluxes at lower altitudes.*

changed

*(17) L18: at the end of this sentence you could again cite the work of Skogsberg as well as Grünewald and Wolfsperger).*

done

*References*

*P15, l21: The year of publication should be placed at the end.*

Done

*Figures*

*Fig.1: In the caption please specify whether DGPS measurements are indicated by*

*the red lines.*

Not all the red lines show DGPS data, in case recent LiDAR DEMs have been available, these have been analyzed. The red lines thus denote survey profiles (either DGPS/DEM or DEM/DEM comparison). This note was added to the caption.

*Fig.3 and following: it is not clear what you mean with "surface elevation*

*changes plotted for surface elevation in 2007" ? Do you mean the difference 2007 –*

*1999 and 2015-2007 ? Please clarify in the captions and also in the ordinate label.*

The Figures have been shifted to the supplement. Y axis has not been changed for the various surveys, every point is displayed with it's elevation in the year 2007. Otherwise, if every point had been displayed with the surface elevation during the surveys, we would compare different locations.

*Fig3.: It shoud be 25th /75th percentile (and not %!)*

Changed

*The captions of all following figures could be reduced:::there is a lot of redundant*

*information.*

As this is part of the supplementary material now, we decided to keep the full caption, as the main manuscript is much shorter and straight forward.

*Fig.13 and 18: on the right subplot "mbm" and "ref" is missing as label*

The indices "mbm" and "ref" are added to the corresponding profiles (Figures shifted to supplementary material and numbers changed).

*References*

*Joint Committee for Guides in Measurements (JCGM): Evaluation of*

*measurement data – Guide to expression of uncertainty in measurement,*

**Kommentar [MST1]:** Wurde das hinzugefügt?

*JCGM 100:2008, GUM 1995 with minor corrections, available at:*

*http://www.bipm.org/utils/common/documents/jcgm/JCGM_100_2008_E.pdf, 2008.*

---

## Author Comment (AC4) · 14 Jul 2016

Dear Chris Stokes, we greatly appreciate the helpful comments, time and effort of the three reviewers. According to their suggestions, we reorganized and rephrased large parts of the manuscript and worked on the figures. We implemented a better description of uncertainties, improved the analysis of the data (presenting an overview instead of all the results at the specific sites), and shifted the results at the specific sites to the supplementary material. In addition to that, the suggested changes in the specific comments were done, and some additional work on grammer, spelling, and smaller corrections. We hope that all who contributed their time now find a good 'response' on this investment - we are grateful for the input and tried our best! Best

regards, Andrea Fischer

---

## Author Response (AR1)

**Response to Reviewers**
**TC-2016-61**
**Local reduction of decadal glacier thickness loss through mass balance management in ski resorts**

A. Fischer[1], K. Helfricht[1] and Martin Stocker-Waldhuber[1]

[1]Institute for Interdisciplinary Mountain Research, Austrian Academy of Sciences, Innsbruck, 6020, Austria
*Correspondence to*: Andrea Fischer (andrea.fischer@oeaw.ac.at)

We thank all the reviewers and the editor for spending their time and effort for providing comments, which are much appreciated. According to the reviewer suggestions, we reorganized and rephrased large parts of the manuscript (see the attached change track version of the final manuscript) and worked on the figures. We implemented a better description of uncertainties, improved the analysis of the data (presenting an overview instead of all the results at the specific sites), and shifted the results at the specific sites to the supplementary material. In addition to that, the suggested changes in the specific comments were done, and some additional work on grammar, spelling, and smaller corrections.

This file is organized in the following way:
- Response to the anonymous reviewer 1
- Response to the anonymous reviewer 2
- Response to the comments of Mauri Pelto
- Change track version of the final manuscript

Response to Anonymous Referee #1

General comments :

*This paper deals with the impact of mass balance management on glacier thickness changes in ski resorts. From photogrammetry, laser scanning and GPS measurements, the authors compared the thickness changes on profiles with and without mass balance measurements over the last 20 years. The authors conclude that thickness changes could be reduced by 35-65% thanks to the mass balance management. This paper shows a large dataset given that 16 profiles on 5 glaciers have been measured since 1997 or 1999. These comparisons are rare on the alpine glaciers and these results certainly deserve to be published. It does not concern the scientific community only but also many people involved in the mass balance management.*

*However, this manuscript has large weaknesses and did not reach a sufficient maturity. This manuscript is difficult to read and confusing.*

*First, the authors should revise the structure: -*

We restructured the manuscript as suggested by the reviewer, extended the descriptions of the data, added more details on measurement accuracies and added a more detailed description of the method.

*Data: a lot of information should be included in Data and not elsewhere in the paper: for instance, the information related to the uncertainties on photogrammetry, GPS.., measurements given in Discussion (lines 5-15 p 10) should be reported in Data section. The authors should check that, everywhere in the manuscript. Seven lines in "Surface elevation data" are not sufficient to describe the measurements given these data are the basis of the paper. The authors should explain here clearly that DGPS measurements of 2014/2015 are compared to DEM from 1997/1999 and 2006/2007. It is not obvious at this stage of the mansucript.*

*-The techniques of management on each glaciers should be summarized in a Table (maybe in the Table 2).*

done

*-In Data and Methods section, the explanations about the emergence velocities (p.5, lines 1-20) should be removed from Data and Methods: first, the authors do not provide any explanations here why and how they used these equations. At this stage, the reader wonders why the authors introduce these Equations relative to the emergence/submergence velocities. These equations should be moved to the Discussion (lines 16-29, p10) where the authors provide a discussion about the relationship between the surface mass balance and the elevation changes.*

*However, I am not sure these Equations are helpful given the authors do not use them. In any case, the authors should use the classical way to present the equation related to emergence velocity (Cuffey and Paterson, 2010, equation 8.65). Equations 2 and 3 are not necessary in any case, given these equations are not used for calculations in this paper.*

The equations of Cogley et al. were replaced by the suggested equation in Cuffey and Paterson. As suggested by reviewer #1, this part was shifted to the discussion.

*-Study sites: the authors should replace the long (and indigestible) description by a Table.*

The Table was improved. The detailed description went to the supplement and were described by a shorter and more illustrative descriptions of the sites.

*-Results: this section is indigestible. The reader does not need the full and detailed description of elevation changes at each pylons, skilifts, pistes. Here, the description seems to come directly from a technical report. It is not useful for the scientific community.*

This part was shifted to the supplement.

*The number of Figures which show the elevation changes (Fig 3, 4, 56,7,8, 9, 10, 11, 12, 13, 14, 15, 16, 17, 18, 19, 20, 21, 22, 23,24 and 25 !!) should be considerably reduced and most of them should be moved in a supplementary material.*

Done

*For this section, the authors should make a strong effort to sum up the results, to analyze them and to make a new Figure to show the summarized results.*

We now present three figures summarizing the results of all sites and all periods. Only one figure presents the types of measurements was kept as an example.

*From my point of view, it is absolutely necessary and this kind of Figure would be useful for the scientific community.*

*Second, the analysis of results is poor. I am aware of the difficulties given that the data come from different techniques and different areas. In this way, it is very difficult to compare elevation changes for areas with different altitudes and different aspect.*

We revised the presentation of the results to give a better overview by adding mean elevations of the test sites, by calculating annual values with the respective error bars and by separating sites with higher and lower thickness losses.

*However, despite on Table 3, there is a lack of quantitative results. I think that Table 3 is not sufficient to analyze the results. In addition, I am not sure that results given in relative reduction % are relevant.*

We removed the relative reduction from the Table.

*Some figures in Table 3 seem to me strange or wrong: for instance, at ST2, the authors reported a relative reduction of -396% despite on the fact that the elevation change is +0.2 m for the reference profile and -0.7 m for the profile with mass balance management.*

The mass balance management mainly influenced surface elevation changes at this location in period 2. We tried to clarify this point in the introduction: during the first period (~1997 to 2006), mass balance management was applied only after 2003. During the second period, mass balance management was applied continuously. This explains why the area mbm in period 1 shows higher thickness losses than the reference area, but lower ones in the period 2. The explanation was in the text, but very well hidden – we apologize and hope that the explanation in the introduction makes it easier to follow the interpretation. We only interpret the differences between period 1 and 2, and the differences between mbm and ref areas in period 2. We removed the relative numbers, as dividing by zero results in odd values.

*Did I miss something? If not, the authors should check the whole results. Again, I do not think the relative reduction in % is meaningfull.*

We removed that.

*When the value of elevation changes are close to zero (close to ELA), the relative reduction can reach very large values but it does not mean that the impact is more important. This way of presenting the results is not convincing. I believe that the percentages given in the manuscript (and in Abstract) are easy to understand for the general public but are probably no relevant.*

We replaced them by absolute numbers and added mean values of thickness change by Abermann et al 2010 for comparison.

*Third, I am not convinced by the conclusions relative to the impacts of mbm. For instance, the authors claimed that "the submergence and emergence should be similar so that a large impact resulting from different or changing ice flow regimes is unlikely". It can be questioned from the results shown in this study.*

Here we have a small misunderstanding, which we hoped to improve by rephrasing the text. We have to assume that submergence and emergence at mbm and ref profiles is similar and not changing too much with time. Otherwise, one could argue that the investigated thickness changes are not resulting from the mass balance management, but from ice flow dynamics. We now show in the study that the shape of the reduced thickness changes exactly fits to the covers for Schaufelferner. We think that this is a good indication that these sharp and rectangular bumps are not caused by changes in ice dynamics.

*For instance, it seems very difficult to make conclusions about the impact of mass balance management when the measurements have been done at very different altitudes (Fig. 3, Fig 5, Fig. 7.) and for different aspects.*

Our approach is to compare areas with mass balance management to areas without mass balance management at various altitudes. High elevations show low thickness changes, low elevations high

thickness without mass balance management. With mass balance management, thickness loss is reduced in all elevations, but at different rates. We agree that modeling the effects of mass balance management would be difficult, as the full energy balance is needed and the course of ablation and accumulation during season can result in huge differences. Nevertheless, we think that this is a further step, but not the aim of this study: We wanted to show that there are effects by a relative comparison.

*Moreover, I do not understand how the submergence/emergence velocities spatial distribution can be neglected in this study.*

We investigate thickness changes, not mass balance, as this is the parameter which ski resorts are most sensitive or vulnerable. Horizontal flow velocities on Austrian glaciers are a few meters/year only, so that the differences in vertical flow velocities between the first and the second period and mbm and ref profiles located only few meters apart should be small. At least there is no known proof of rapid velocity changes between the first and the second period, and no indications of extremely changing flux divergence at the profiles. This should be evident from cracks and crevasses. In addition to that, pylons mounted at the glacier surface would have to be repositioned in case of such an event, leading to an official report on that event.

A reduction of ice flow velocity at the glacier tongues lead to increased thickness loss even at constant melt rates. A partially reduction of thickness loss in mbm areas at glacier tongues caused by changes in ice flow would be related to increasing flow velocities, which is not observed.

*The authors wrote that "Interannual differences in emergence/submergence velocities are less than 0.5 m a-1 at Kesselwandferner", but, here, this is the spatial distribution of emergence velocities which is questioned. Or I missed something.*

We need booth assumptions for our study. For Kesselwandferner, we actually measured emergence velocity of stakes separated only a few meters. The differences in emergence or submergence are small, unless ice flow velocities are ~100 m/year and the stake is located in a crevasse zone. This type of motion is clearly indicated by surface features as crevasses, and therefore we can exclude that. We did not include that in the discussion, as this topic is far from the main focus of the paper.

*The results shown TCD in Fig 3 to 25 are confusing and again, a thorough analysis and a synthesis are missing to provide relevant results and to convince the reader.*

We added the summary Figures and shifted the raw data figures to the supplement.

*Many things should be improved but I think it is not necessary to make a list at this stage given that the structure of the manuscript and the analysis of the results should be strongly revised first. Detail information should be removed from the manuscript when there are not used in the manuscript (GPR data, history of ski tourism). The authors should check that carefully.*

We removed the section on GPR data, but kept the evolution of glacier ski resorts, because it would not be quite straight forward to understand why the infrastructure is located at the current positions now causing the need for adaptation (at least for someone which is not too familiar with length and thickness changes on Eastern Alpine glaciers in last 40 years).

Response to the *anonymous referee #2*

General Comments

*The paper presents a valuable, comprehensive and comprehensible overview about the medium-term (decadal) effect of technical modifications of the glacier surface mass balance within Austrian Skiing resorts. The application of these measures started around the year 2004 and the related physical processes and short-term effects were already investigated in detail in a number of earlier studies. The authors analyze digital elevation model differences as well as DGPS measurements at selected spots of different glaciers with and without application of such measures between multiple years in order to quantify the effect of these intentional modifications on surface elevation changes within this timescale. Results indicate the clear medium-term benefit as well as the limitations of these technical measures on a larger scale in terms of costs and efforts.*

*Although the uncertainty of their method is discussed in the manuscript, the latter should be done in a more thorough, quantitative way, thereby also using an appropriate and exact terminology. In a revised version of the manuscript, the individual uncertainty sources should not only be named but all of them also be estimated and Interactive the resultant combined expanded uncertainty as well as its impact on the main results comment of the paper calculated.*

We restructured the manuscript and added thorough calculation of measurement uncertainties as well as a better description of other uncertainties.

*Therefore I suggest accepting the paper after the points listed in the specific comments and some minor ones in the technical corrections have been implemented by the authors.*

*Specific Comments (in decreasing order of importance)*

(1) *In the discussion section (p10. Lines 5-15) the Authors indicate a maximum uncertainty for their method of 1.1 m for both the DGPS and the DEM differences. It is not clear a) how this number is calculated exactly (uncertainty components), b) what confidence interval it is referred to (e.g. standard (66%) or expanded (95% level) uncertainty), c) what the impact of the combined expanded uncertainty is on the main results of the paper. For clarity and consistency, I very much encourage the Authors to study and use the Guide to the Expression of Uncertainty in Measurement (GUM; JCGM, 2008)) as well as the terminology that is defined therein.*

The discussion of errors and uncertainties was shifted to the section on Data and methods and expanded, discussing systematic and measurement errors separately.

*(2) It is not clear how areas with long-term mass balance management were exactly identified (onsite location) in the study (own (GPS) records or data from skiing resorts?). Please add this information.*

The criteria for the selection of the test sites are now part of the paragraph on test sites. The locations have been clearly indicated by previous (own) GPS records and documents of the pre-projects. Of course only own DGPS measurements are presented here.

*(3) Concerning the single effect of grooming on snow and ice ablation, the authors should add that the observed effect was in the order of only 5 % rather than 10% and that this number was very close to the measurement uncertainty (Olefs and Fischer, 2008; Fischer et al., 2011; ;Olefs, 2005;Olefs and Obleitner, 2007).*

Done – we changed the number to the original 6% and added the information that this is close to the measurement accuracy.

*It is also worth to clarify the following in the paper: Based on previous studies, it is still not clear what exact physical mechanism(s) leads to the observed effect. Beside the reduction of surface layer erodibility through compaction (stronger bonding of the snow crystals), there may be other effects, e.g. a modification of surface albedo due to a reduction of average grain size of the surface snow layers induced by the snow-cat or a modification of snow thermal conductivity (Olefs and Obleitner, 2007). If there are new studies that separate those exact effects on the ablation reduction known to the authors, they should cite them.*

We changed the wording to distinguish measurements and unknowns.

*(4) I strongly suggest adding units (SI) to all variables whenever formulas or variables are used in the manuscript (e.g. p.5).*

The formula is replaced.

*(5) The physical effect of water injection in the snow cover is mainly to add mass to the existing seasonal snow (if there is enough cold content in the snow to refreeze the injected water). After injection, the release of latent heat due to refreezing of the water decreases the absolute value of the cold content of the existing snow cover (as e.g. shown in Fig.7 of Olefs and Fischer, 2008). Firstly, I do not understand why the cold content should be increased by this method (p.3, line 19).*

*Secondly, the authors could also add the two main resulting limitations of this method apart from the large effort: enough cold content before injection and timing problem (enough time between applications).*

We agree with that points (cold content can increase only if small amounts of water are injected and cold air can penetrate the snow cover through the holes for a longer time; no skiing and no grooming and no snow falls after the injection). As injection is not used as mass balance management method, but mainly for ski racing, we decided for shortening that part rather than going to deep into that topic.

*Technical corrections*

*p1 (1) l30: ":::depth height:::??"*

height removed

a, J Glac

*p2 (3) l6: please explain the first occurrence of the shortcut "GI" (4)*

done

1986

Section rephrased

*(6) l18 and others: I would prefer "t-bar lift" instead of "tow lift" throughout the paper*

replaced

*p3 (7) l19: increase or decrease cold content? you could use the absolute value to clarify::.*

Water injection is not discussed any more

*p4 (8) l19: please use consistent naming for "Austrian glacier inventory" (GI?)*

ok

The section is removed

The section is removed

*P7 (11) L20: here the single effect of grooming (compaction of the surface layers) is mixed with the potential effect of snow farming (lateral transport of snow mass by snow cats), please clarify.*

The section was rewritten. In any case, we can not separate the effects of various measures in this spatially distributed study.

changed

changed

changed

*(15) L12: "Fujita and Ageta, 2000" is missing in the references*

added

*(16)L16: I would suggest to write "(e.g. without glacier cover)" as a) in other regions of the world glaciers do exist at low altitudes and b) the fact that the effectiveness of surface textiles to reduce ablation decreases with altitude is not tied to the surface type (glacier or not) but it is due to the Interactive energy balance being dominated by sensible/latent heat fluxes at lower altitudes.*

changed

*(17) L18: at the end of this sentence you could again cite the work of Skogsberg as well as Grünewald and Wolfsperger).*

done

*References*

*P15, l21: The year of publication should be placed at the end.*

Done

*Figures*

*Fig.1: In the caption please specify whether DGPS measurements are indicated by*

*the red lines.*

Not all the red lines show DGPS data, in case recent LiDAR DEMs have been available, these have been analyzed. The red lines thus denote survey profiles (either DGPS/DEM or DEM/DEM comparison). This note was added to the caption.

*Fig.3 and following: it is not clear what you mean with "surface elevation*

*changes plotted for surface elevation in 2007" ? Do you mean the difference 2007 –*

*1999 and 2015-2007 ? Please clarify in the captions and also in the ordinate label.*

The Figures have been shifted to the supplement. Y axis has not been changed for the various surveys, every point is displayed with its elevation in the year 2007. Otherwise, if every point had been displayed with the surface elevation during the surveys, we would compare different locations.

*Fig3.: It shoud be 25th /75th percentile (and not %!)*

Changed

*The captions of all following figures could be reduced:::there is a lot of redundant*

*information.*

As this is part of the supplementary material now, we decided to keep the full caption, as the main manuscript is much shorter and straight forward.

*Fig.13 and 18: on the right subplot "mbm" and "ref" is missing as label*

The indices "mbm" and "ref" are added to the corresponding profiles (Figures shifted to supplementary material and numbers changed).

Unfortunately, Keller et al. do not find empirical data on the effect of grooming on snow albedo. Measurements of this effect might be difficult for various reasons, amongst them:

- Bidirectional reflection of snow requires careful consideration of all combinations of grooming tracks and incidence angles
- During operation, ski tracks replace grooming tracks, with even more difficult to capture changes in optical properties.

[Figure]

Figure 1: A single skier changes surface albedo on 26.05.2005 in 2850 m.

Taking into account, that grooming during the investigated period ended with mid to end of May, and surface melt processes had been observed mainly from beginning of May onwards, the effect of grooming on albedo during melting season is thus small, which was the reason for skipping that topic initially.

*2-1: to store and maintain snow:::.*

changed

*2-13: The to They*

changed

*2-17: Not only has visitor demand developed over time but cable car technology has advanced::.*

changed

*3-3: Crevassed reduced not just at ski areas but on other glaciers too, for example Colgan et al (2016) Pelto and Hedlund (2001).*

We agree, this is stated also for example in Fischer, A. (2010) Glaciers and climate change: Interpretation of 50 years of direct mass balance of Hintereisferner, Global and Planetary Change 71, 1-2: 13-26.

*3-5: Is removal of rock, sand and dirt from the piste not a goal? Grooming and new snow production oth increase the albedo. This is a goal noted by some of your previous research.*

These measures have been included in the first point which comprises measures to decrease surface roughness. We added explicit examples to that point. The albedo topic is included in the third point, reduction of mass loss, as the high albedo is not an aim on its own purpose, but for its influence on mass balance. But as stated before, grooming during ablation season is rare.

*7-26: I assume the 35% and 65% reduction are compared to adjacent areas of the same glaciers, if so more clearly state this. Somewhere it would helpful to reference typical thickness loss values from either WGMS reporting Austrian glaciers or from the inventory, as a wider reference.*

We restructured the article including a better description of the reference measurements. Typical losses from the glacier inventory have been added.

Abermann, J., Lambrecht, A., Fischer, A., and Kuhn, M.: Quantifying changes and trends in glacier area and volume in the Austrian Ötztal Alps (1969-1997-2006), The Cryosphere, 3, 205-215, doi:10.5194/tc-3-205-2009, 2009.

The thickness of Ötztal glaciers reduced by 0.95 m/year in average between 1969 and 1997, and -0.91 m/year between 1997 and 2006.

*9-2: Continuous grooming will increase albedo.*

This section was restructured. The albedo discussion was included in the state of the art report. Unfortunately we do not have data on surface albedo.

*10-26: I agree with this assertion "In any case, submergence and emergence should be similar for the profiles and the reference profiles"*

We rephrased this part.

*11-2: Grooming would also reduce albedo.*

We added the information that grooming takes place during winter, and the effect is measured in summer, so that a direct influence of grooming on albedo is not very likely.

*12-3: It is worth noting that mass balance management extends to Tignes, France; Whistler, BC and Mount Hood, OR.*

We added this information together with other ski resorts from our personal knowledge.

*12-28: The enhanced prominence of managed area versus managed areas, generates steeper slopes as noted. This in turn should increase ablation. Will also act both as a wind scour and potentially wind trap for accumulation. Is this observed?*

Yes, both effects are observed, but not quantified in detail. The removal of snow from previously covered platforms leads to their rapid meltdown once the maintenance stops. The separation of radiative and wind drift effects is hard to measure and drawing general conclusions included high uncertainties.

*Figure 1: Ski area boundary line should be more distinct color.*

changed

*Figures: The number of profile figures is impressive. However, collectively they are redundant and also detract from highlighting important overall trends. The variation from profile to profile becomes the focus. I would suggest utilizing only two sets from each glacier, or focusing more on the central panel. The central panel alternative takes TCD advantage of the fact that Table 3 provides the data from the third panel for each glacier.*

We restructured the Figures.

*Figure 1 provides profile location. Hence, you could just use the middle panel for all but two profiles on each glacier.*

Figure 1 is changed, and we hope that you find it benfical.

[revised manuscript text omitted]

spatial resolution of 1x1 m. The spatial resolution can be considered similar to the DGPS measurements (acquired every second at walking velocity resulting in a point density of 1-2points per meter XXXXX). The DEMs of the second glacier inventory have been processed in a 20x20 m grid. This grid was then resampled to a 5x5 m grid. As tehsethese have been recorded before mass balance management started, the glacier surface has been smooth, so that the deviations within one pixel from the pixel mean result from the surface slope, which is lower than 20 ° in the test sites. Therfore, the difference of point altitudes from the mean within the grid cells is lower than 1.3 m for slopes with 20 ° and lower than 0.08 m for slopes with 5°.

All the elevation data was recorded during ablation season, with major parts of the glacier showing bare ice. As every year and every region show a different course of accumulation and ablation, it is not possible to survey each site at it's individual minimum of mass. Although thus minor impacts of old snow (from the winter) can not be excluded, all data were recorded in absence of new snow. Therefore, the effect of seasonal snow cover is neglected. In any case, the time span of one decade includes considerable interannual variability in glaciological and meteorological parameters. The effect of old snow, confined to highest elevations at the survey dates, is lower than this the year to year variability, as the early date of the survey in 2015 coincides with extreme melt rates and early recession of snow cover.

**2.3 The s**

**2.52.3 Study sites**

During 2003-2009, areas with mass balance management on Stubai Glacier, Pitztal Glacier, Kaunertal Glacier and Sölden Glacier ski resort have been monitored extensively, with at least two surveys per anno and a maximum of weekly surveys on Stubai glacier. After finishing these projects, the sites were still monitored on an annual basis with few ablation stakes and a photographic documentation of the evolution of the glacier surface. Based on this documentation, areas with continuous mass balance measured within these ski resorts have been selected for this study. Although not being subject of the initial research projects 2003-2009, sites in the Hintertuxer glacier ski resort have been included in this study, as these are the sites with the longest history of mass balance management by on-glacier snow production. Mass balance management takes place in areas where technical infrastructure located on solid ground borders to ski pistes on glacier parts with high subsidence rates, at pylons on glacier or boarder parks with jumps, jibs and pipes. Exemplarily, three locations with mass balance management are shown in Figure 2. The middle station at Schaufelferner (Figure 2 a) is located on a rock, with the surrounding glacier showing high subsidence rates. Glacier covers are applied since 2004 to allow the access to and exit from the station. The steepening tongue of Rettenbachferner is kept in shape with a combination of snow production and covers to provide easy access to the valley station, from which the photography in Figure 2 b) is taken. The subsidence of glacier surface is highest at the tongues, but also takes place in highest elevations. Most striking 
[revised manuscript text omitted]
, thus, have a large range between the 25th and the 75th percentile of all surface elevation changes. The profile SOE1a is located at the lowest elevated part of the glacier tongue. It shows the largest spread between mean annual surface elevation changes of both periods in the *mbm* area, while mean annual surface elevation losses in the *ref* area were nearly constant.

For the second category, interpretation of surface elevation changes at these profiles is more complex with respect to their location on the glaciers. Thickness losses in both managed and unmanaged area are with a mean of - 0.4 m/y and -0.6m/y in period 1 
[revised manuscript text omitted]

---

## Author Response (AR2)

We thank the reviewer and the editor for their valuable comments, which are very much appreciated. We went through general and detailed comments. Unfortunately, they referred to the change track version, which was very confusing as large parts of the manuscript have been redone. This was the reason why the final work in the previous submission was done on the 'changes accepted' version. This was intended to state in the comments to the editor, but obviously got lost somewhere. Unfortunately we did not communicate this in the response of the authors. We are sorry for the reviewer and the editor that they spent their time on a pre-final version. Most of the comments refer to mistakes which had already been corrected, or text parts/formula which have been removed in the final version.

List of relevant changes:

We rewrote the sections Discussion and Conclusion, skipped Figure 7, skipped the profiles of the third category, extended the description of the categories, renamed all Figures/Captions/Profiles/Tables as suggested. Therefore, the numbering of the remaining profiles changed (Short comment in the supplementary file, SOE3a-> SOE3, KT5-> KT1).The suggested detailed changes were done, and some minor additional corrections.

General comments of reviewer:
      Authors response

*1) Compared to the first version, this manuscript has been largely improved.*
*The presentation of data and results is clearer. However, the manuscript needs to be improved again and did not reach the sufficient maturity to be published in TC. Many points need to be clarified. Many sentences are obscure and need to be reformulated (see general and specific comments). Some words are missing. Some words are not English. There are mistakes in some Equations. The authors did not take time to reread thoroughly the revised manuscript.*
      We made final corrections and proof reading in the 'changes accepted version', as large parts of the manuscript were reorganized and we wanted to avoid the situation that incomplete sentences etc. are left in the final manuscript. This was based on the opinion that the change track version would be used for change tracking only, not as basis of the review.

I have the feeling to play the role of colleagues or co-authors to prepare a draft and make a submitted version. Although there are large improvements compared to the first version which was a kind of "technical report", my feeling is that the authors did not take sufficient time to revise the first version. Could the authors be fair and take pity on Reviewers ?
      We are sorry for the misunderstanding which version of the manuscript was used for review.

In any case, this manuscript deserves a publication in TC after the necessary following changes.

2) The results section is not clear enough although that I recognize it has been largely improved. The presentation of the 3 categories is unclear: which difference? Information about the first and the second category is needed. The captions of Fig 5 and Fig 6 are exactly the same and are not helpful. The authors wrote "in similar settings" (l. 12, p. 11) but it is very unclear.

Line 12 on page 11 change track version says:
'profiles close to the glacier terminus in similar topographic settings resulting at similar thickness loss at mass '
Is this referring to this text or another?

This is line 12 page 9 in the final version.

What we want to say is that in the same climatic regime, same exposure, same accumulation condition, same radiation conditions, the mass balance of two points at the glacier neighboring each other (few meters to tens of meters distance) should not differ significantly, as we see no driver for this difference in terms of accumulation or energy balance governing melt. Can we basically agree on the absence of sharp changes in mass balance fields, in absence of any topographic, radiative or hydrologic cause for sharp gradients? (Here I consider wind erosion as a topographic factor).

We tried to improve the description of the categories, and skipped the third category as suggested.

3) Results section: Did the authors remove every measurements where the ice has disappeared between the first and the second measurements?

As the authors surveyed the profiles in situ, during ablation season, the existence of ice in the profiles is certain.

In Figure 4, one can see that the elevation change close to the terminus (altitude close 2880 m, light blue) for the first period, is not very negative (the values are between -5 and 0). It could be due to the fact that the ice has disappeared before the end of the 1st period.

We agree that it makes no sense to investigate profiles without ice cover. This was never intended, nor done.

 If it is the case, the results are biased and the elevation changes are not relevant. It is absolutely necessary to remove these values (Table 3 and Figure 4) to make a relevant

analysis.

It is not entirely clear to me on which evidence the hypothesis of ice free profiles is based on. It is correct that the largest elevation changes in the Austrian glacier inventories are found close to glacier tongues (cf Kuhn et al., 2012). Not every glacier in fact does show very high elevation changes, and not at every part of the tongue. We have no indication that there is any error in the data, or that any parts are ice free, not even after the summer of 2015.

4) About the results on the third category: The elevation changes for these small features are very heterogeneous. I do not think such data are useful for this paper because the representativeness is strongly questioned. I think that the authors should remove Figure 7 and data of this third category in Table 3.

The intention was to show which effect the coverage of kickers and bumpers has. These features have become modern in the last decades, and originally needed loads of winter snow to be built. This is not the case any more, as on normal pistes the ground now is shaped, and on glaciers the shape is preserved with covers. We agree that this is not very relevant for this study, and skipped the third category.

5) Important: the results are confusing and the reader is lost because the authors wrote everywhere in the manuscript "with mass balance management" for the first period although there is not "mass balance management" during this period except the last years of this period (l. 25, p.4). The authors should reword it because it is very very confusing. For instance, the authors could use "reference areas" and "experimental areas", and specify "experimental areas without management" for the first period and "experimental areas with management" for the second period....or something else. Check carefully the manuscript. Same thing in captions of Figure 5 and 6. In addition avoid deltaZmbm for the first period in captions of Fig 5 and 6. Change the name of the vertical axis in Fig 5 and 6. Very misleading. (by the way, Figures 5 and 6 are fine and it is a large improvement compared to the first version).

We hoped that it would be clarified by the detailed description, that only part of the first period is covered by mass balance management. Nevertheless, we changed the wording as suggested and hope that the article is less confusing.

This comment refers to line 23 page 4 in the final version. As in this sentence only the areas are described, the exp and ref are introduces later and changed in the whole manuscript and supplement.

7) L. 21-31, p. 12: This part should be rewritten or removed. It seems to come from a technical report with advices to ski resort managers.
We removed that part.

8) Discussion about the surface mass balance inferred from elevation changes: it is not convincing.
We did not aim at inferring mass balance from elevation change. This is stated more clearly in the introduction now:
'From this basic research question, this study aimed at assessing long-term net effects of mass balance measures on volume changes by comparing surface elevation changes in areas which have been subject to different types of mass balance management and neighbouring areas without such management. This study is not aiming at inferring mass balance from elevation changes.'
A whole section in the discussion is now used, with the help of the corrected formula, that we have too much uncertainties to infer mass balance from thickness changes.

The Discussion about this point is poor. The assumption relative to "In any case, submergence and emergence should be similar for the profiles and the reference profiles" (l. 22, p. 16) is not supported by any data or evidence.
That is true, as on only one glacier in Austria (Kesselwandferner) time series of ice flow velocities are measured. So the ski resort glaciers are not subject to measurements of emergence and submergence. We skipped that sentence. It was actually wrong, as we use that as a precondition to compare thickness changes in adjacent profiles with very low flow velocities.

In this Discussion, the Equations are not used.

There is only one equation? We now refer to the equation.

I think that the authors should do a thorough analysis (first, please, write Equation of Cuffey and Paterson correctly ).
First: Done

The authors should use this Equation to calculate the uncertainties on mass balance derived from elevation changes, between periods 1 and 2.

We did not derive any mass balance, and we do not think that this makes any sense or adds some information, as we do not know u, v, or w. Without knowing ice thickness, density in the vertical columns and 3D ice flow velocity the uncertainty of inferring local mass balances from elevation changes is too high to make sense. Comparing local stake measurements of ablation from the previous project with thickness changes, we know that at that time (2004-2009) about 60 to 70 % of the melted ice was replaced by inflowing ice, with approx.. 4 m of ice melt resulting in 1 m thickness loss.

I think they should only use data with similar elevation changes during the first period (see my previous comment) for such analysis. Please remove the other equations which are not helpful (in addition, Equation named (1) is probably wrong given the density is not the same

for the surface mass balance and the entire column and the surface mass balance (deltam).

There is no equation 1 nor 2?

Very confusing. Please remove the equations named (1) and (2) (the authors forgot to name Equation of Cuffey and Paterson). Discussion should be reworded completely.

The discussion is reworded. There is only one equation left.

9) The section "Conclusions" is not well written and should be reworded completely.

done

10) Table 3: did the authors remove every values for which the ice disappeared between the 1st and the 2nd measurements ?
There were no values to remove, as we did not include ice free areas.

11) Figure 3: It should be useful to show the "management areas" with dashed line (first panel). a) and b) should improve Figure 3 and caption. The coordinates are not necessary and the authors should add an horizontal scale. In caption, the authors should mention the name of the glacier (Stubai)
We decided not to show a dashed line, as then the sharp gradient in elevation change is not visible. We think that the margins of the mass balance management are clearly visible in the hillshade (b).
Stubai indeed is the name of a mountain range, or a glacier ski resort, and not a glacier. Table 1 and Figure 1 should explain that.

12) Figure 4: Provide a), b) and c) for each panel and provide full explanations in the caption about the colors (light blue, dark blue, pink, red)
done

13) Figures 5 and 6: In the caption, the authors used "median". Do the authors mean "median " or "mean" ? Change the vertical axis according my previous comments. Revise the captions according my previous comments
'Median' is intended to mean 'median', otherwise it would be phrased as 'mean'. Axis and captions changed.

14) Remove Figure 7. Not helpful for the manuscript.
done

15) I believe that the English of the manuscript should be checked by a native speaker. Many sentences seem to me obscure (although I am not native speaker).
The manuscript was checked by our English editorial office, Dr. Scott. As she is a professional language editor (although mostly working with literature), grammar and spelling should be ok. She had been going to this revision also.

Specific comments:
The number of lines correspond to the revised version with track changes

a) Check the ref Abermann et al (p. 2). Year ?
2009

b) Check the brackets for Smiraglia et al (p. 2)
this was corrected in the final version

c) L. 20, p. 2: "opened" ? I do not understand
changed to 'started operation''
d) L. 2 and 3, p.3: unclear
'The transition of the ski tracks from glacier to the bare ground changes constantly with variations in glacier surface altitude and snout position.'
Reworded to:
The transition of the ski tracks from glacier to the bare ground changes constantly, as the glacier area and elevation changes.

e) L 6 p 3: unclear
Where glacier ice has disappeared, bare ground is often steeper than and not as smooth as the former glacier surface, so that pistes have had to be rerouted to meet the requirements on width and difficulty.

We splitted the sentence.

f) L. 18 p 4: confusing
Although the short-term effect has been proven, it could be that measures have not been applied frequently enough to return a sustainable result, or that ice dynamics lead to a redistribution of masses so that, for example, no effect on surface elevations would be measurable.
Changed to
After a decade of measuring the glaciers, the question arises of the long-term outcome of these measures: Although the short-term effect has been proven, it could be that measures have not been applied frequently enough to return a sustainable result., or thatAlso ice dynamics can be considered to lead to a redistribution of masses so that, for example, no effect on surface elevations would be measurable.

g) L. 16-17, p 6: how did the authors check the accuracy ? It largely depend on the distance to reference station, to the time of acquisition and to number of satellites. The authors should add these information. Where does the standard deviation come from ? Vague

The 'Magnet Tools' software of TOPCON allows to calculate standard deviation, as any other GPS Software. The software is given in the line above. We added it once again.

h) Check the numbering of Equations
done, numbers removed (makes no sense using two equations).

i) L. 9, p 7: remove this Equation. Unuseful
done

j) L. 23-26 p 7: very confusing. Explain how you obtained the uncertainties
The line above says: 'The measurement errors of the thickness changes at one location are the sum of the measurements errors of each surface elevation data set' and the respective numbers are given in section 2.1.

k) P. 7 : some words are missing, check it. Please Re-read carefully the manuscript.
We and our editor carefully checked the manuscript (final version, not change track), but did not find too much errors – maybe this was a problem in the wrong version?

l) L. 2-3 p 8: reformulate please
The DEMs of the second glacier inventory have been processed in a 20x20 m grid. This grid was then resampled to a 5x5 m grid.
Rephrased to
The DEMs of the second glacier inventory were processed in a 20x20 m grid and resampled to a 5x5 m grid.

m) L.9-15: the text should be revised thoroughly. Finally the authors do not mention the real uncertainty on elevation changes (after interpolation). Very misleading.
Is this referring to page 8?
'All the elevation data was recorded during ablation season, with major parts of the glacier showing bare ice. As every year and every region show a different course of accumulation and ablation, it is not possible to survey each site at it's individual minimum of mass. Although thus minor impacts of old snow (from the winter) can not be excluded, all data were recorded in absence of new snow. Therefore, the effect of seasonal snow cover is neglected. In any case, the time span of one decade includes considerable interannual variability in glaciological and meteorological parameters. The effect of old snow, confined to highest elevations at the survey dates, is lower than this the year to year variability, as the early date of the survey in 2015 coincides with extreme melt rates and early recession of snow cover.'

We agree that the estimation of errors in elevation models is complex, but we are uncertain about the idea of a 'real uncertainty'. In lines 9-15, no interpolation is mentioned?

n) "Study sites " section: "high subsidence rates": not clear. Where does this result come from ? Reference ? Relevant in this section ?

We found two sentences with 'high subsidence rates':
   i)      Mass balance management takes place in areas where technical infrastructure
           located on solid ground is adjacent to ski pistes on glacier parts with high

subsidence rates, and at pylons on glacier or boarder parks with jumps and pipes.

ii)     The middle station at Schaufelferner (Figure 2 a) is located on a rock, with the surrounding glacier showing high subsidence rates.

Both describe why we investigated these locations. We consider this information relevant.

o) L. 29 p 8: this glacier is not mentioned anywhere (except in Figure 1 in very small characters) and the reader is lost…
The glacier is mentioned in Table 1, also displaying which glacier belongs to which ski resort.

p) L. 33 p 8: which striking effect ? not relevant in this section

q) P. 9 words are missing.

r) L. 12, p 9: check the verb
In the five glacier ski resorts (Table 1), 24 sites with mass balance management have been selected for comparing thickness changes in managed and reference areas. The comparison was carries out for two time periods for each profile.

This was corrected already  in the final version.

s) L. 29 p. 17: per year ?
yes

t) Please Reword Discussion and Conclusions (see my previous general comments)
done

[revised manuscript text omitted]

To highlight why we do not claim to interpret thickness changes in terms of mass balance, we shortly [BS1]refer to basic glaciology: Much more importantly, sSurface elevation changes $\frac{\partial s}{\partial t}$ result from glacier dynamics, density ($\rho$) changes and point mass balance $b$ (Cuffey and Paterson, 2010).

$$\frac{\partial S}{\partial t} = \frac{b}{\rho} + w - u\frac{\partial S}{\partial x} - uv\frac{\partial S}{\partial x}$$

In our study, we measured sSurface elevation changes $\frac{\partial s}{\partial t}$ at reference and experimental areas. All other parameters are unknown, and modelling results at such a small scale may be dominated by uncertainties of input parameters.

A necessary assumption for being able to compare $\frac{\partial s}{\partial t}$ for experimental and reference areas with obvious and photographically documented differences in mass balance b is that the other components ($\rho, w, u\frac{\partial s}{\partial x}, v\frac{\partial s}{\partial x}$) are largely unchanged do not change much between experimental and reference areas. Therefore, For our analysis, we have to presume that the components of the surface velocity at the point *u*, *v*, and *w* are similar in the mass balance managedexperimental and the reference profiles, as well as over nd during the two periods. Then, the measured thickness changes are driven by the measures and not by glacier dynamics. This hypothesis assumption is not confirmed by measurements of mass balance, densities or flow velocities, but at least supported by some empirical evidence:is confirmed by i) the superficialafical forms at mass balance managed areas evidence and are stable both 
[revised manuscript text omitted]